# Structural analysis of Red1 as a conserved scaffold of the RNA-targeting MTREC/PAXT complex

Anne-Emmanuelle Foucher [1,6], Leila Touat-Todeschini [2,6], Ariadna B. Juarez-Martinez [1,6], Auriane Rakitch[2], Hamida Laroussi[1], Claire Karczewski[2], Samira Acajjaoui[3], Montserrat Soler-López[3], Stephen Cusack [4], Cameron D. Mackereth [5] ✉, André Verdel [2] ✉ & Jan Kadlec [1] ✉

To eliminate specific or aberrant transcripts, eukaryotes use nuclear RNA-targeting complexes that deliver them to the exosome for degradation. *S. pombe* MTREC, and its human counterpart PAXT, are key players in this mechanism but inner workings of these complexes are not understood in sufficient detail. Here, we present an NMR structure of an MTREC scaffold protein Red1 helix-turn-helix domain bound to the Iss10 N-terminus and show this interaction is required for proper cellular growth and meiotic mRNA degradation. We also report a crystal structure of a Red1-Ars2 complex explaining mutually exclusive interactions of hARS2 with various ED/EGEI/L motif-possessing RNA regulators, including hZFC3H1 of PAXT, hFLASH or hNCBP3. Finally, we show that both Red1 and hZFC3H1 homo-dimerize via their coiled-coil regions indicating that MTREC and PAXT likely function as dimers. Our results, combining structures of three Red1 interfaces with in vivo studies, provide mechanistic insights into conserved features of MTREC/PAXT architecture.

An astonishing amount of RNA, including a sizeable portion of aberrant transcripts, is constantly being synthesized within eukaryotic nuclei[1–3]. To detect and eliminate specific or aberrant transcripts, cells rely on highly regulated RNA degradation machineries. Unwanted transcripts are recognized by specialized RNA-targeting complexes that deliver them to the nuclear RNA exosome for degradation[4,5].

In the fission yeast *Schizosaccharomyces pombe*, MTREC (Mtl1–Red1 core) or NURS (nuclear RNA silencing) is such an RNA-targeting complex and is responsible for recognition of cryptic unstable transcripts (CUTs), meiotic mRNAs and unspliced pre-mRNAs[6–8]. The MTREC core module comprises the zinc-finger protein Red1 and the RNA helicase Mtl1. In addition, there are four associated sub-modules - the nuclear cap-binding complex (Cbc-Ars2), the Iss10-Mmi1 complex that targets meiotic RNAs, the Pab2-Rmn1-Red5 complex likely recognizing poly(A) sequences and the more loosely associated poly(A) polymerase Pla1[6–8]. The subunit conservation indicates that MTREC is the fission yeast counterpart to the human PAXT (pA-tail-exosome-targeting) complex that targets long and polyadenylated nuclear RNAs[7,9–11]. Indeed, PAXT contains a large zinc-finger protein ZFC3H1 and the RNA helicase MTR4, human homologs of MTREC Red1 and Mtl1, as well as the cap-binding and poly(A)-binding sub-modules[9–11]. The well-studied human NEXT (nuclear

[1]Univ. Grenoble Alpes, CNRS, CEA, IBS, F-38000 Grenoble, France. [2]Institut for Advanced Biosciences, UMR Inserm U1209/CNRS 5309/University Grenoble Alpes, La Tronche, France. [3]Structural Biology Group, European Synchrotron Radiation Facility (ESRF), CS 40220, 38043 Grenoble, France. [4]European Molecular Biology Laboratory, 71 Avenue des Martyrs, CS 90181, Grenoble Cedex 9 38042, France. [5]Univ. Bordeaux, Inserm U1212, CNRS UMR 5320, ARNA Laboratory, Institut Européen de Chimie et Biologie, 33607 Pessac, France. [6]These authors contributed equally: Anne-Emmanuelle Foucher, Leila Touat-Todeschini, Ariadna B. Juarez-Martinez. ✉e-mail: cameron.mackereth@inserm.fr; andre.verdel@univ-grenoble-alpes.fr; jan.kadlec@ibs.fr

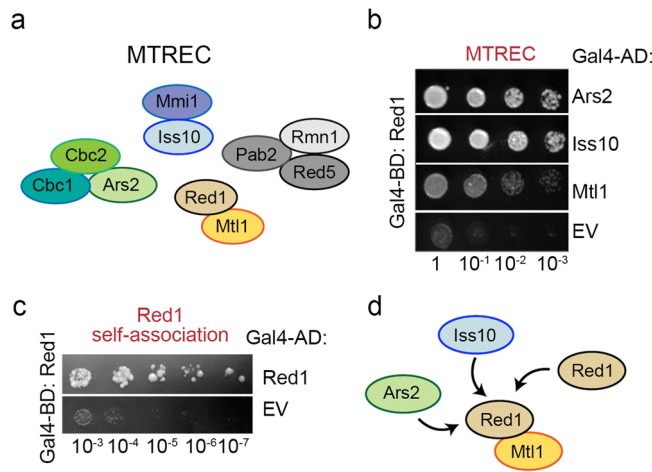

**Fig. 1 | Red1 forms multiple interactions within the MTREC complex.**
**a** Schematic representation of the MTREC complex. Red1 and Mtl1 form the MTREC complex core[7]. **b** Y2H assays revealed that Red1 interacts with Ars2 and Iss10. Mtl1 was used a positive control. Red1 was used as bait. Growth assayed on selective medium lacking histidine and with 50 mM 3-amino-1,2,4-triazole (3AT) histidine inhibitor is shown. EV- empty vector, BD- binding domain, AD- activating domain. **c** Y2H analysis of Red1 self-association. Growth was assayed on medium lacking histidine and with 30 mM 3AT. **d** Schematic representation of the interactions characterized in this study.

exosome-targeting) complex, on the other hand, is composed of the MTR4 helicase, zinc-knuckle protein ZCCHC8 and the RNA-binding subunit RBM7[12]. NEXT is involved in degradation of short-lived RNAs, such as promoter upstream transcripts (PROMPTs) and enhancer RNAs (eRNAs)[12,13] but its relation to MTREC is less clear.

The subunit composition of the *S. pombe* MTREC and human PAXT complexes has been characterized by proteomics[6–9,11]. The Red1 protein was shown to form a scaffold of the MTREC complex, interacting with multiple subunits and linking individual sub-modules together[14]. Red1 consists of 712 residues (compared to the 1989 residues of its ZFC3H1 counterpart in human PAXT) and is predicted to be mostly disordered. A crystal structure of a minimal Red1-Mtl1 complex, revealed how Red1 interacts with the Mtl1 Arch domain via its zinc-finger containing domain at its C-terminus[14]. In human and *S. cerevisiae*, Mtl1 counterpart MTR4 has been structurally characterized[15] revealing molecular details of its interactions with its partners including the NEXT subunit ZCCHC8[16] and the RNA exosome to whom MTR4 presents the targeted RNA for degradation[17,18]. Recently, using yeast two-hybrid (Y2H) and pull-down assays Dobrev et al. defined the minimal interacting regions involved in the interactions of Red1 with Iss10 of the Iss10-Mmi1 sub-module and with Ars2 of the cap-binding complex[14]. However, there is no 3D structural information available on the interactions that the Red1-Mtl1 core forms with the individual MTREC sub-modules. Similarly, the function of these modules in MTREC/PAXT activity still remains unclear.

The MTREC complex is involved in specific elimination of meiotic transcripts and heterochromatin formation in *S. pombe* and this activity involves the Iss10/Mmi1 sub-module[6–8,19,20]. Indeed, in *S. pombe*, expression of a number of meiotic genes is strictly repressed during the vegetative growth by selective RNA degradation, since their aberrant expression is highly deleterious to cell viability[21]. Meiotic transcripts and non-coding RNAs that contain repeats of a hexanucleotide motif called determinant of selective removal (DSR) are specifically recognized by the YTH family RNA-binding protein Mmi1, which in turn recruits MTREC to mediate their degradation[20–24]. In vegetatively growing cells, Mmi1 forms nuclear foci[21], where most of its partners, including MTREC, nuclear exosome and the target RNA, co-localize[6,8,25]. The Iss10 subunit is also directly involved in the selective

elimination of meiotic DSR-containing transcripts and is important for the MTREC sub-nuclear localization[26]. In addition, Iss10 and the MTREC-dependent RNA degradation has been connected to the mTOR cellular energy-sensing signaling[27]. Under vegetative growth, Iss10 is phosphorylated by TORC1 (TOR complex 1), which prevents its degradation by the proteasome. However, under conditions of nitrogen depletion, TORC1 phosphorylation of Iss10 is blocked, and Iss10, in turn, eliminated[27]. Lack of Iss10 comes with a loss of MTREC co-localization with Mmi1 in nuclear foci, and accumulation of DSR-containing mRNAs and meiotic proteins, thereby allowing the cells to proceed with meiosis[26,27]. Thus, Iss10 is a key regulatory element of MTREC-dependent degradation of meiotic mRNAs, that most likely links the MTREC core complex to the Mmi1 bound RNAs[26].

The Red1-Mtl1 core also recruits the Cbc-Ars2 sub-module[6,7], which has been more extensively studied in human. CBC (nuclear cap-binding complex) is a key RNA biogenesis regulator that co-transcriptionally binds the m7G cap at the 5′ end of RNA polymerase II transcripts[28,29]. The CBC and its partner ARS2 are implicated in fate determination of Pol II transcripts, including their processing, transport or degradation[30]. CBC-ARS2 interacts in a mutually exclusive way with positive RNA biogenesis factors (such as PHAX, FLASH and NCBP3), or with the RNA destructive MTREC/PAXT and NEXT assemblies[30–32]. While the connection to PAXT and NEXT involves ZC3H18[9], no such protein has been identified in the MTREC complex[6,7]. The atomic details underlying how the CBC-ARS2 complex is recruited to the exosome-linked RNA targeting complexes and what role this module plays are poorly understood.

Here, we report a biochemical and structural analysis of two important interactions of Red1 with MTREC sub-modules, combined with in vivo studies. We determined an NMR structure of the Red1-Iss10 complex, which reveals how Red1 recognizes Iss10 via its helix-turn-helix motif. We also report a crystal structure of the Red1-Ars2 complex, which shows how the Ars2 C-terminal zinc-finger domain interacts with the conserved EDGEI motif of Red1. In addition, we demonstrate equivalent interaction between ZFC3H1 and hARS2 within the PAXT complex, providing structural insights into how the Cbc-Ars2 complex is integrated into the RNA-targeting complexes. Functional in vivo experiments using Red1 structure-driven mutants showed that both Iss10 and Ars2 interactions are important for MTREC-mediated cellular activities. Finally, we demonstrate that Red1 dimerizes via its coiled-coil region, strongly suggesting that the MTREC complex acts as a dimeric assembly and this feature is conserved in human PAXT and NEXT complexes.

## Results

### Yeast two-hybrid analysis of Red1 interactions within MTREC

Red1, together with the Mtl1 helicase, has been shown to form the MTREC core[6,7,14] (Fig. 1a), but the structural basis of its interactions with the MTREC sub-modules remains elusive. To understand how Red1 associates with its partners, we first performed Y2H assays to analyze its interactions with several MTREC members. In agreement with the recent Y2H analysis of the MTREC complex interactions[14], we detected strong interactions with Ars2, Iss10 and Mtl1 as well as between Iss10 and Mmi1 (Fig. 1b and Supplementary Fig. 1a, b). In addition, our Y2H assay revealed an unexpected Red1 self-association (Fig. 1c). To obtain mechanistic insights into the MTREC/PAXT complex architecture, in this study, we performed structural and functional characterization of the Red1 interactions with Iss10 and Ars2 as well as of its self-association (Fig. 1d).

### NMR structure of the Red1-Iss10 complex

A Y2H and GST pull-down analysis recently revealed that Red1[187–236] and Iss10[1–51] represent the interacting regions of the complex Red1 forms with Iss10[14]. Our characterization of the complex using Y2H, Strep-tag pull-down, and limited proteolysis experiments identified equivalent

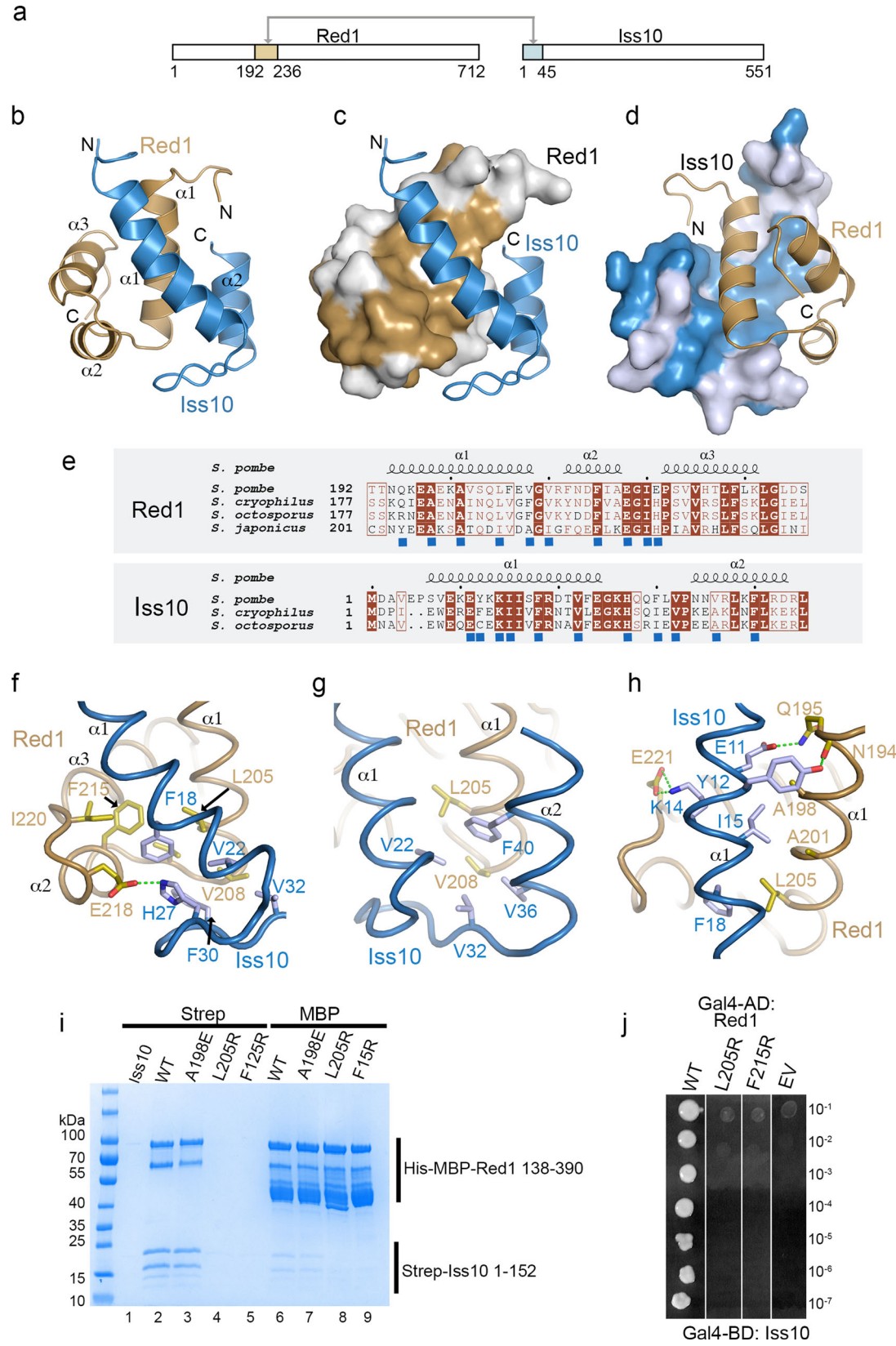

fragments Red1[192–236] and Iss10[1–45] (Supplementary Fig. 1c–e). Multi angle laser light scattering (MALLS) showed that these constructs form a homogenous complex sample with a 1:1 stoichiometry (Supplementary Fig. 1f).

The Red1[192–236]·Iss10[1–45] complex was then subject to structural analysis (Fig. 2a). While the complex resisted crystallization, we obtained a high quality [15]N-HSQC NMR spectrum (Supplementary Fig. 1g) that prompted us to determine its structure by NMR. The resulting structural ensemble (Supplementary Table 1; Supplementary Fig. 2a, b) revealed that Red1 residues 192–236 and Iss10 residues 1–44 comprise the structured core of the interaction complex. Red1[192–236] folds into three short helices arranged in a way reminiscent

**Fig. 2 | Red1 interacts with Iss10 with its HTH domain. a** A schematic diagram highlighting the Red1 and Iss10 interacting regions. **b** Ribbon representation of the lowest energy NMR-based structure of the Red1-Iss10 complex. The Red1 HTH domain is shown in brown and the Iss10 N-terminus is in blue. The secondary structures are labeled. **c** Surface representation of Red1 with conserved surface residues shown in brown. **d** Surface representation of Iss10 with conserved surface residues highlighted in blue. **e** Sequence alignment of the Red1 HTH domain (upper panel) and Iss10 (lower panel) in different *Schizosaccharomyces* species. Identical residues are in brown boxes. Blue squares indicate residues involved in the interaction. **f** Details of the interaction between the Iss10 helix α1 and the following loop with Red1 helices α1 and α3. **g** Details of the Red1 helix α1 packing against the Iss10 surface made of helices α1 and α2. **h** Coiled-coil interaction and hydrogen bonding between Red1 and Iss10. **i** Strep-tag pull-down experiments with Strep-Iss10[1–45] and His-MBP-Red1[192–236] mutants indicated above the lanes. Iss10[1–45] and Red1[192–236] were co-expressed, the cultures were split into two halves and the proteins purified on Strep-Tactin or amylose resins. In absence of the interaction with Red1, Strep-Iss10[1–45] is unstable. Source data are provided as a Source Data file. **j** Essential role of Red1 L205 and F215 in the interaction with Iss10 shown by Y2H assay. Growth was assayed on medium lacking histidine and with 30 mM 3AT.

to the classical helix-turn-helix (HTH) motif (Fig. 2b–d and Supplementary Fig. 2c). Indeed, a search of the Protein Data Bank using the PDBeFold server at the EBI revealed that Red1[192–236] is highly similar to other HTH motifs, such as the DNA binding domain of the *B. subtilis* DeoR protein[33] (Supplementary Fig. 2c–e). Nevertheless, the surface of Red1 helix α3, that in classical HTH motifs frequently mediates dsDNA binding, carries mostly hydrophobic residues (Supplementary Fig. 2f). While Red1 DNA binding has not been reported and no unspecific DNA binding was observed during Red1 purification, specific DNA interaction cannot be excluded. The Iss10[1–45] fragment consists of two helices, packing tightly against Red1 (Fig. 2b–d). Mutual interaction surfaces on both proteins are made of residues that are highly conserved among evolutionary distant *Schizosaccharomyces* species underlining the importance of this interaction (Fig. 2c–e).

### The Red1-Iss10 interface

The interface between Red1 and Iss10 is dominated by hydrophobic interactions. Iss10 helix α1 and the following loop form numerous contacts with the groove between Red1 helices α1 and α3 (Fig. 2f). Iss10 F18, V22, F30, and V32 interact with the Red1 hydrophobic surface composed of L205, V208, V210, F215 and I220. E218 also makes a salt bridge with H27 (Fig. 2f). In parallel, Red1 helix α1 packs against the Iss10 surface made of helices α1 and α2. Red1 L205 and V208 interact with Iss10 residues around V22, V32, V36 and F40 (Fig. 2g). Finally, Iss10 and Red1 establish a short coiled-coil interaction, including contacts between Red1 A198, A201 and L205 and Iss10 I15 and F18 (Fig. 2h). These interactions are further strengthened by several hydrogen bonds of Iss10 E11, Y12 and K14 and Red1 N194, Q195 and E221 (Fig. 2h).

In order to assess the role of the Red1-Iss10 interaction in vivo, we were then interested in identifying Red1 mutations that could disrupt its interaction with Iss10. To this end, we generated an A198E mutation in the center of the coiled-coil formed with Iss10 and two L205R and F215R mutations that we hypothesized should directly disrupt the interaction with Iss10 helix α1 (Fig. 2f–h). These mutations were introduced into the His-MBP-Red1[138–390] and tested for their interaction with Iss10[1–152] by using Strep- and MBP- pull-down assays (Fig. 2i). While A198E retained the WT level of binding (Fig. 2i, lanes 3 and 7), the L205R and F215R mutants no longer bound Iss10 (Fig. 2i, lanes 4, 5, 8 and 9). Since co-expression with Red1 is required for Iss10[1–152] stability, Iss10[1–152] produced with these Red1 mutants becomes unstable and does not purify employing the Strep-Tactin resin. To assess the possible impact of the L205R and F215R mutations on the Red1 HTH domain, we produced Red1[192–236], but 1D NMR analysis revealed that in absence of Iss10, the WT and mutated variants are unfolded (Supplementary Fig. 2g). Together with size exclusion chromatography, the 1D NMR spectra confirmed that both L205R and F215R mutants are similar in behavior to WT Red1[192–236] being soluble with no sign of aggregation or higher order oligomerization (Supplementary Fig. 2g, h). We then confirmed the essential role of L205 and F215 for the interaction with Iss10 in the context of full-length proteins using an Y2H assay (Fig. 2j). Ultimately, the L205R mutation was selected for in vivo studies (see below).

### Red1-Iss10 binding interface is required for proper degradation of meiotic DSR-containing mRNAs and clustering of Red1 foci in vivo

In order to evaluate in vivo the function of the Red1-Iss10 binding interface, recombinant Red1-TAP or Red1-L205R-TAP proteins in combination with Iss10-GFP protein were expressed in *S. pombe* cells in place of the endogenous proteins from the respective endogenous promoters. Using these cells, reciprocal co-immunoprecipitation experiments showed that the Red1-L205R mutation also compromised Red1 association with Iss10 in *S. pombe* (Fig. 3a and Supplementary Fig. 3a). Next, we examined the growth of *red1-L205R* mutant cells. While *red1Δ* cells showed only a moderate growth defect on solid rich medium at 30 °C, this defect was more pronounced at lower temperature of 25 or 18 °C (Supplementary Fig. 3b), as published previously[14,34] or when the cells were grown on minimal medium (Supplementary Fig. 3c). Importantly, *red1-L205R* cells showed a similar growth defect, when grown on minimal medium, suggesting that Red1-Iss10 interaction can be important for Red1-dependent cell growth function depending on the environmental conditions (Fig. 3b). We then examined the subcellular localization of Red1-L205R-TAP mutant and Iss10-GFP. Both Red1 and Iss10 have been previously reported to co-localize within nuclear foci[23,26,34]. We observed that the *red1-L205R* mutation had an effect on both Iss10 and Red1 localization by, respectively, inducing the disappearance of Iss10 nuclear foci and the increase in the number of Red1 foci per nucleus (Fig. 3c–e). These findings indicate that the interaction of Iss10 with Red1 is required for Iss10 recruitment to Red1 nuclear foci and suggest that it may promote the clustering of Red1 nuclear foci. Finally, we examined the level of meiotic DSR-containing mRNAs which are degraded by the MTREC machinery[7]. Expression of the Red1-L205R mutant, instead of WT Red1, led to a defect comparable to *iss10Δ* mutation, and it was more pronounced when cells were grown on minimal medium (Fig. 3f and Supplementary Fig. 3d). Taken together, we conclude that in vivo, the Red1-Iss10 binding interface plays a role in mediating functions of Red1, and its importance is determined by the environmental growth conditions.

### Crystal structure of the Ars2-Red1 complex

The interaction between Ars2 and Red1 has recently been shown, by Y2H and pull-down assays, to be mediated by a short conserved "EDGEI" motif of Red1[14]. Using Y2H and gel filtration chromatography we confirmed the implication of the EDGEI motif in the interaction with full-length Ars2, identifying a slightly longer construct Red1[20–40] (Fig. 4a and Supplementary Fig. 4). To further characterize the interaction between Ars2 and this Red1 motif, we used isothermal titration calorimetry (ITC). Both Red1[1–54] and Red1[20–40] (with and without MBP tag) bound Ars2 with an equivalent dissociation constant ($K_d$) of about 5 μM (Fig. 4b and Supplementary Fig. 5a, b), confirming that additional sequences surrounding EDGEI motif do not enhance the binding affinity for Ars2.

*S. pombe* Ars2 consists of 609 residues and in analogy to hARS2 is predicted to contain intrinsically disordered N- and C-termini and a central structured region. Ars2 residues 450–516 were shown to bind Red1 by Y2H[14], but in our hands short constructs encompassing this

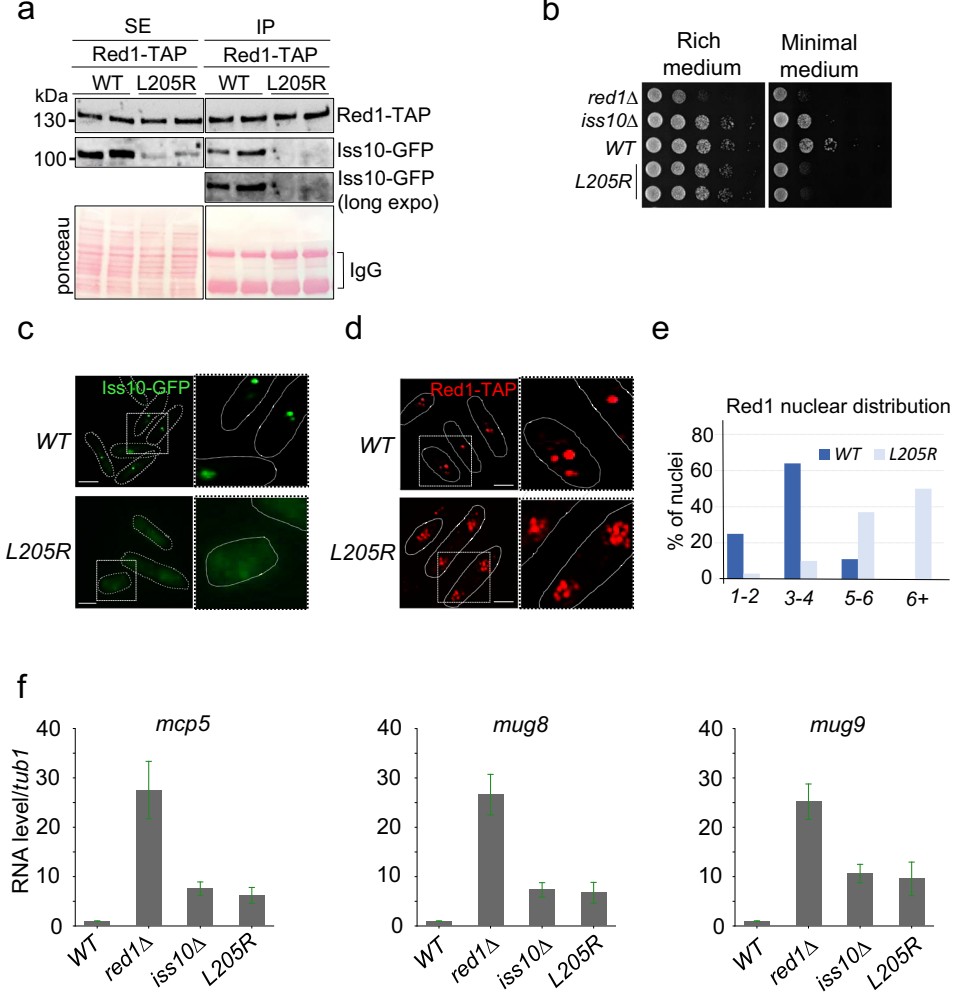

**Fig. 3 | Red1^{L205R} mutation affects cell growth, degradation of DSR-containing mRNAs and clustering of Red1 foci. a** Co-immunoprecipitation experiments, analyzed by western blot, showing that Red1^{L205R} mutation compromises Red1-Iss10 interaction in vivo. Soluble cell extracts (SE) were prepared from exponentially growing cells expressing Red1-TAP and Iss10-GFP in liquid minimal medium. **b** Plating assays monitoring the growth of of *red1-L205R* compared to *WT, Iss10Δ* and *red1Δ* cells. Cells were grown on solid rich or minimal media for 5 days at 25 °C. **c** Images of live fluorescence microscopy showing the localization Iss10-GFP in *WT* and *red1-L205R* cells. Scale bar = 5 μm. **d** Images of immuno-fluorescence microscopy showing Red1-TAP localization in *WT* and *red1-L205R* cells. Scale bar = 5 μm. **e** Graph showing the number of Red1 foci per nucleus in *WT* and *red1-L205R* cells. A minimum of 100 nuclei from two different biological isolates were counted. **f** Quantitative RT-PCR experiments showing the level of *mcp5, mug8* and *mug9*, three meiotic and DSR-containing mRNAs normalized to *tub1* mRNA in *WT, red1Δ, iss10Δ* and *red1-L205R* cells. Total RNAs were purified from cells growing exponentially in liquid minimal medium. Error bars represent standard deviation from three independent biological replicates. Source data are provided as a Source Data file.

region of hARS2 did not produce stable protein. Based on deletion mutagenesis, limited proteolysis and sequence comparisons to hARS2, we generated the Ars2^{107–527} construct lacking both disordered termini, which turned to be more stable than the full-length protein and retained its interaction with Red1. The strength of the interaction depends on salt concentration, since when NaCl concentration was reduced from 200 to 100 mM, the $K_d$ measured by ITC for Ars2^{107–527} and Red1^{20–40} increased from 5 to 0.8 μM (Fig. 4c).

To gain molecular insights into the Ars2-Red1 interaction, we set out to determine the atomic structure of the complex by X-ray crystallography. First, we obtained crystals of the complex between Ars2^{107–527} and Red1^{20–40}, that however only diffracted to 5 Å. Inspired by the recently available AlphaFold2 model of *S. pombe* Ars2 (AF-094326)[35], we then extended both the N- and C-terminal limits of the Ars2^{107–527} construct and removed one predicted mobile loop (residues 184–205). This new construct was produced by co-expression of its two fragments - Ars2^{68–183} and Ars2^{206–531}. Two separate elution peaks of Ars2 were obtained by size exclusion chromatography (Supplementary Fig 5c, d) and MALLS analysis identified them as a homodimer and

a monomer, respectively (Supplementary Fig. 5e, f). Only the minor dimeric version of Ars2 in complex with Red1^{20–40} yielded single crystals, which diffracted to a 2.8 Å resolution. The structure was determined by molecular replacement using the AlphaFold2 Ars2 model (AF-094326) and refined to an $R_{free}$ of 30.2% and an $R_{work}$ of 24.5% (Supplementary Table 2).

The *S. pombe* Ars2 structure consists of four defined domains, including a helical core with an inserted RNA recognition motif (RRM) and is flanked by two extended, mostly helical structures. These extensions are referred to as the N- and C-terminal leg, in similarity to its human counterpart[32] (Fig. 4d–f). Ars2^{68–183,206–531} crystallized as a domain-swapped dimer, having the C-terminal leg segment exchanged between the two protomers (Fig. 4e). The hinge region, allowing the rotation of the C-terminal leg, corresponds to residues 420–421 (Supplementary Fig. 7a). Analysis of the extensive dimer interface with PDBePISA server[36] at the EBI revealed that it buries 4946 Å² on each Ars2 protomer, including 37 hydrogen bond and 18 salt bridge contacts between the two molecules and has a Complex Formation Significance Score (CSS) of 1, indicating that the interface plays an

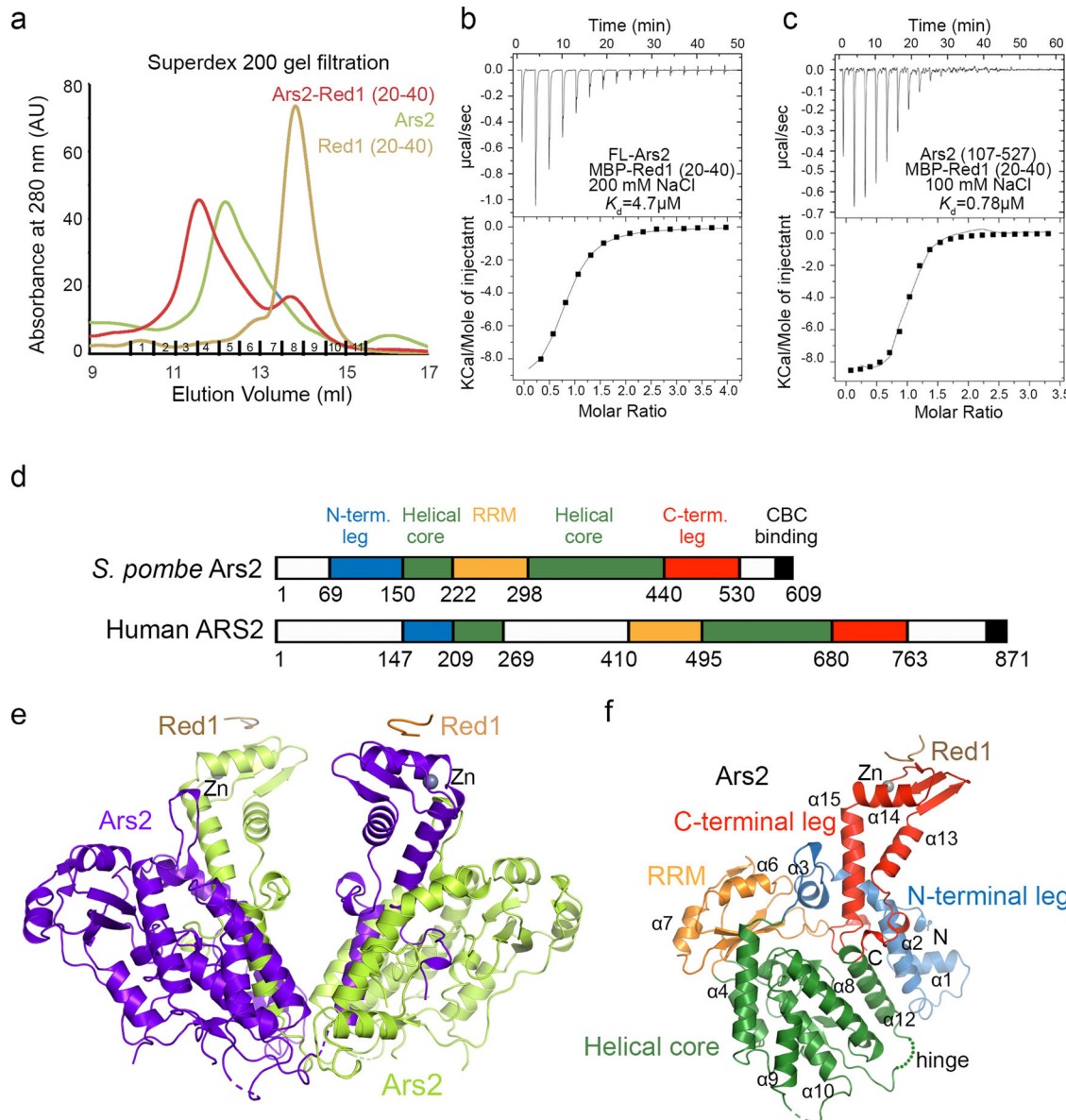

**Fig. 4 | Crystal structure of the Ars2-Red1. a** Overlay of Superdex 200 gel filtration elution profiles of full-length (FL) His-MBP-Ars2, His-MBP-Red1[20–40] and their complex. The proteins were expressed individually as MBP fusions and purified on amylose resin. Both proteins were first purified on Superdex 200. **b** ITC measurement of the interaction affinity between FL Ars2 and His-MBP-Red1[20–40] in presence of 200 mM NaCl. **c** ITC measurement of the interaction affinity between Ars2[107–527] and His-MBP-Red1[20–40] in presence of 100 mM NaCl. **d** Schematic representation of the domain structure of *S. pombe* and human Ars2, as defined in this study and in Schulze et al.[32]. The CBC binding site was characterized in[43]. **e** Ribbon

representation of the overall structure of the Ars2[68–183,206–531] dimer. The C-terminal region (420–531) corresponding to the C-terminal leg is swapped between the protomers. The interacting peptide of Red1 is shown in brown. **f** Crystal structure of the Ars2 monomer with C-terminal leg not swapped in complex with Red1. The four defined domains are highlighted in different colors. Red1 peptide is in brown. The hinge region, allowing for the mobility of the C-terminal domain is shown. Annotated secondary-structure elements correspond to those shown in the alignment of Supplementary Fig. 6.

essential role in the dimeric complex formation. The domain swap occurred likely already during the overexpression in bacteria as it could not be reproduced even with high concentrations (up to 40 mg/ml) of the monomeric Ars2. Since it is not known, whether such an Ars2 dimer has a physiological significance, in the following text, the monomeric Ars2 structure, without the domain swap, will be considered.

**The Ars2-Red1 interface**
Red1 interacts with the Zn-finger domain of the Ars2 C-terminal leg, particularly with the surface formed by α14, β11 and β12 (Fig. 5a). In our Ars2-Red1 complex structure, electron density is visible for six out of the twenty Red1 peptide residues (Supplementary Fig. 7b). The peptide

adopts a U-shape structure bent around the central G34 and establishes hydrophobic and charged contacts with Ars2, burying 300 Å[2] of its surface. Red1 G34 and I36 together with aliphatic portions of D33 and E35 side chains pack against the Ars2 hydrophobic surface generated by K473, L484, F485, L486 and F490 (Fig. 5b). In addition, the three negatively charged residues E32, D33 and E35 make several hydrogen bonds and salt bridge interactions with Ars2 K483, K493, and H494 (Fig. 5c). The electrostatic nature of these interactions is consistent with the variation of the $K_d$ with salt concentration (Fig. 4b, c). Finally, Red1 I36 and G34 form main chain interactions with Ars2 L484 and L486 (Fig. 5c).

In order to identify the key Ars2 and Red1 residues for stabilizing the interaction, we mutated Ars2 K483, which forms salt bridge

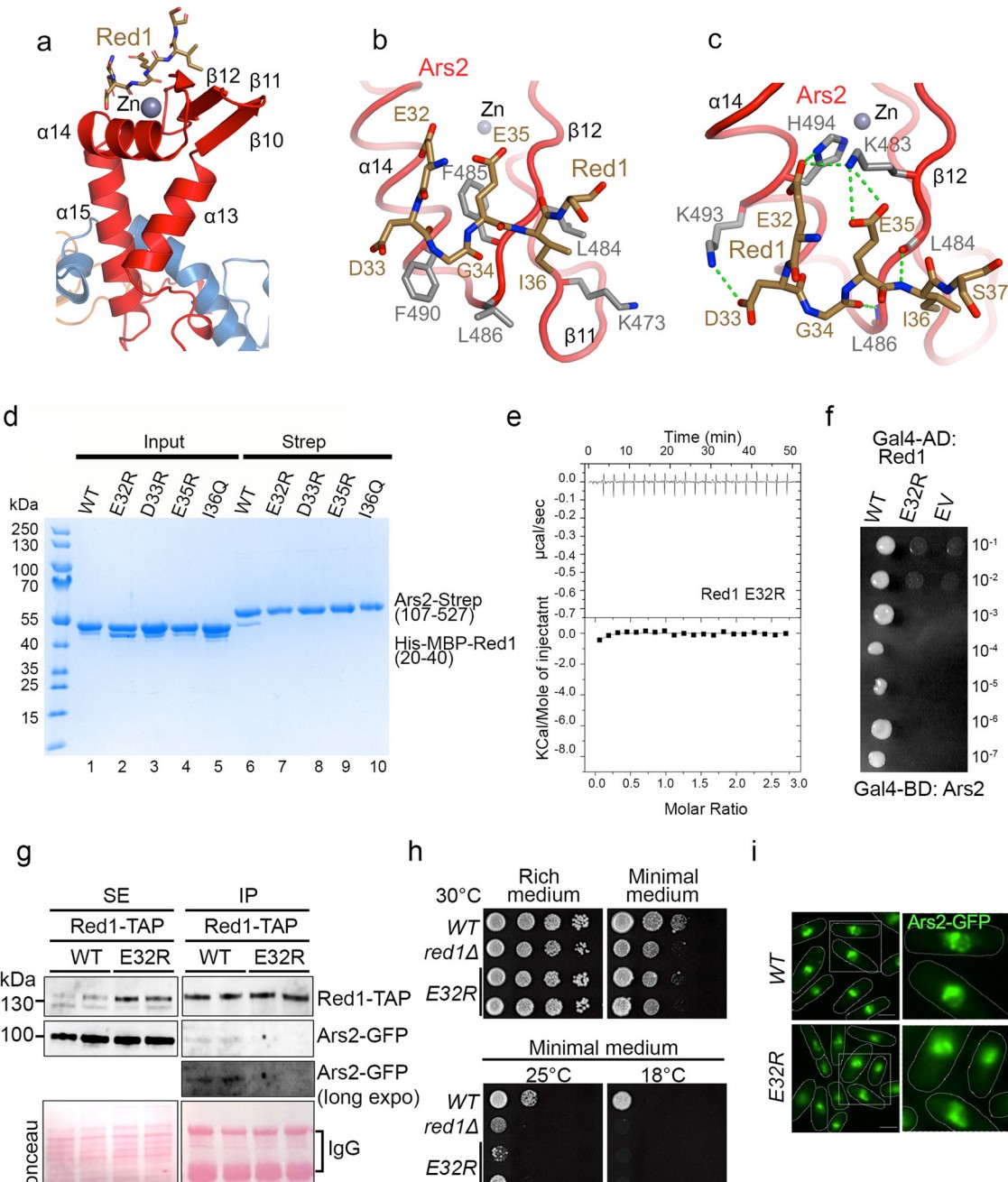

**Fig. 5 | Red1-Ars2 interaction is necessary for proper cell proliferation and Ars2 nuclear localization into Red1 foci. a** The Red1 peptide (in brown) interacts with the Zn-finger domain of the Ars2 C-terminal leg (in red). **b** Details of the interaction between the Red1 EDGEI motif and the Ars2 surface formed by α14, β11 and β12. Hydrophobic contacts between Red1 and the Ars2 residues are shown. **c** Residues of the Red1 EDGEI motif form several side and main chain hydrogen bonds and salt bridge interactions with the Ars2 C-terminus. **d** Strep-tag pull-down experiments with Strep-Ars2[107–527] and His-MBP-Red1[20–40] mutants indicated above the lanes. Ars2 and Red1 proteins were first purified using Ni[2+] resin (lanes 1–5). Ars2 with either WT or mutant Red1 were then co-purified on Strep-tactin resin (lanes 6–10). All mutations abolished the interaction. **e** ITC measurement of the interaction affinity between Strep-Ars2[107–527] and the His-MBP-Red1[20–40] E32R mutant in presence of 100 mM NaCl. **f** Essential role of Red1 E32 in the interaction with Ars2 as shown by Y2H. Growth was assayed on medium lacking histidine and with 30 mM 3AT. **g** Co-immunoprecipitation experiments of Red1-TAP and Ars2-GFP in *WT* and *red1-E32R* cells, analyzed by western blot. Soluble cell (SE) extracts were prepared from exponentially growing cells in minimal liquid medium. **h** Plating assays monitoring the growth of *WT, red1Δ and red1-E32R* cells on rich and minimal media at different temperatures. **i** Images of live cell fluorescence microscopy showing Ars2-GFP localization in *WT* and *red1-E32R* cells. Scale bar = 5 μm. Source data are provided as a Source Data file.

interactions with Red1, as well as F490, which is central to the hydrophobic contacts between the two proteins to aspartates (Fig. 5b, c). The integrity of the Ars2 mutants was verified by gel filtration, with an elution profile similar to the WT protein (Supplementary Fig. 7c). In Strep-tag pull down assays, both mutations (K483D, F490D) essentially disrupted the Red1 binding (Supplementary Fig. 7d, lanes 3, 4). We also mutated key Red1 residues including Glu32, Asp33, Glu35 and Ile36 within the Red1 peptide (20–40). In pull-down assays with Strep-tagged Ars2[107–527], individual mutations of all four residues abolished the binding (Fig. 5d, lanes 7–10), confirming the importance of these residues for the interaction. In addition, the role of E32R was supported by ITC measurements, since no binding to Ars2[107–527] was

observed for the peptide containing the E32R mutation (Fig. 5e). In a previous study, an E32A, D33A, E35A triple mutant was required to disrupt the binding between full length Red1 and Ars2[14]. Using Y2H assays, we could show that the Red1 E32R mutation is sufficient to prevent the interaction with Ars2 in the context of full-length proteins (Fig. 5f). The structure-guided mutagenesis thus validated the key residues observed in the structure for the Red1-Ars2 interaction and identified the Red1 E32R mutation as a suitable candidate for functional analysis.

### In vivo, red1-E32R mutation affects Red1-Ars2 interaction, cell growth and Ars2 localization in nuclear foci

We then examined the function of Red1-Ars2 binding interface in vivo. We assessed the impact of the *red1-E32R* point mutation on Red1-Ars2 interaction by co-immunoprecipitation experiments. Recombinant Red1-TAP, Red1-E32R-TAP and Ars2-GFP proteins were expressed instead of endogenous proteins and from respective endogenous promoters. Importantly, while Red1-TAP and Ars2-GFP interact, as expected, this interaction was lost in cells expressing Red1-E32R-TAP (Fig. 5g). Next, we conducted growth assays on different solid media and at different temperatures. As observed previously, the *red1Δ* cells experience a growth defect, which is exacerbated on minimal medium (Fig. 5h and Supplementary Fig. 3b, c) or at lower temperature (Supplementary Fig. 8a; refs. 14, 34). A similar growth defect was visible for the *red1-E32R-TAP* mutant cells grown on minimal medium, suggesting that Red1-Ars2 interaction, depending on the growth conditions, can be important for the normal growth of the cell population (Fig. 5h). As MTREC degrades a specific population of RNAs such as meiotic DSR-containing mRNAs and PROMPTs/CUTs[7], we asked whether the levels of these transcripts changed in *red1-E32R* mutant cells. However, no significant changes were observed for several meiotic mRNAs and PROMPTs/CUTs MTREC targets (Supplementary Fig. 8b, c). It has been demonstrated that Ars2 regulates *pho1* and *byr2* protein-coding genes[37]. We examined the possible effect of *red1-E32R* mutation on *pho1* and *byr2* mRNAs. However, no significant changes in *pho1* and *byr2* mRNA levels were observed between wild-type and *red1-E32R* mutant cells (Supplementary Fig. 8d). We also assessed the relevance of Red1-Ars2 interaction for their respective localization in nuclear foci. In cells expressing the Red1-E32R-TAP mutant protein, Ars2-GFP signal also formed nuclear foci like in wild-type cells but they tended to be more difficult to distinguish from the more even nuclear GFP signal when compared to wild-type cells (Fig. 5i). Interestingly, quantification of the intensity of the signal within each nucleus showed a reduction of Ars2-GFP nuclear signal in *red1-E32R* mutant cells compared to wild-type cells, suggesting that some of Ars2-GFP proteins diffuse to the cytoplasm (Supplementary Fig. 8e). The localization of Red1-TAP in nuclear foci did not significantly change (Supplementary Fig. 8f, g). Thus, in vivo, Red1-Ars2 binding interface is critical for Red1-Ars2 interaction, for optimal cell growth and nuclear localization of Ars2 where Red1 and other subunits of MTREC localize.

### Comparison of *S. pombe* and human Ars2

Compared to human ARS2, the mutual orientations of the domains in *S. pombe* Ars2 are considerably different. When the two proteins are aligned via their helical domains, large differences in positions of the N- and C-terminal legs and to a lesser extend of the RRM domain are apparent (Fig. 6a, b). Similar differences are also observed when compared to the structure of the *A. thaliana* ARS2 ortholog known as SERRATE[38]. The folds of individual domains, however, are rather well conserved between *S. pombe* and human (Supplementary Fig. 9a–d). Thanks to extending the N- and C-terminal limits of the construct used, the *S. pombe* Ars2 structure shows a more elaborate N-terminal leg, where the extreme N terminus (residues 69–80) packs against helices α1 and α2 on one side and interacts with C-terminal residues 527–530 on the other one (Supplementary Fig. 9e). The remaining part of the

N-terminal leg resembles the one of human ARS2 (Supplementary Fig. 9a). The helical core of Ars2 has a similar topology to its human counterpart (Supplementary Fig. 9b) and has the RRM domain inserted after helix α5. While the RRM domain β-sheet surfaces frequently carry exposed aromatic residues in the two RNP1 and RNP2 motifs that are involved in the binding of ssRNA sequences[39], the Ars2 RRM possesses only one W267 located on β3 (RNP1) (Supplementary Fig. 9c). In addition, the β-sheet and W267 are packed against α11 of the helical core, making it unavailable for interaction with RNA. The C-terminal leg can also be well superimposed onto the human C-terminus (Supplementary Fig. 9d). However, an important difference is the presence of a Zn atom coordinated by C476, C481, H494, and H499 in *S. pombe*, which is lacking in human ARS2, due to replacement of C481 by a serine (Fig. 6c). The Zn co-ordination is preserved in *A. thaliana* ARS2/SERRATE[38]. The mutual orientation of Ars2 domains is unlikely to be affected by the C-terminal leg swap, as observed in our low resolution structure of Ars2[107–527], where no domain swap occurs, while the Zn-finger region and the N-terminal helix α2 is partially disordered (Supplementary Fig. 9f).

### The Red1-Ars2 interface is conserved in human

It has been reported that the EDGEI motif is conserved in the human PAXT subunit ZFC3H1, a putative human counterpart of Red1, implying that an analogous interaction between ARS2 and ZFC3H1 might exist within the PAXT complex[14]. Correspondingly, the Ars2 residues that interact with the Red1 EDGEI motif are generally very well conserved across species (Fig. 6c, d) suggesting that the interaction of Ars2 with this motif could be universal. The only exceptions are *S. pombe* L484 and L486, which are both replaced by lysine in most other species that likely retain hydrophobic contacts with I36 (Fig. 6c, d and Supplementary Fig. 10a). To assess the conservation of the interface in human PAXT, we used pull-down experiments with GST-tagged human ARS2[147–871] and MBP-tagged ZFC3H1[12–33] which revealed a direct interaction between ARS2 and ZFC3H1 mediated by the conserved ZFC3H1[12–33] motif (Fig. 6e, lanes 4 and 7). Importantly, the E23R mutation analogous to the above-described E32R mutation in Red1 (Fig. 5d–f) severely reduced ZFC3H1[12–33] binding to hARS2 in an MBP-pull down assay (Supplementary Fig. 10b, lane 3). An I27L mutation did not perturb the binding (Supplementary Fig. 10b, lane 4), indicating that isoleucine in position 27 is not strictly required and can be replaced by a leucine. We could also show that a short peptide containing only 12 residues (ZFC3H1[20–31]) is sufficient for the interaction with ARS2 (Supplementary Fig. 10b, lane 2). To further confirm the conservation of the Ars2-Red1 interface, we used MBP-pull down assays, where the GST-tagged human ARS2 (residues 147–871) bound equally well to MBP-Red1 or MBP-ZFC3H1 "EDGEI" motif-containing peptides (Supplementary Fig. 10c).

In addition to ZFC3H1, human ARS2 also interacts via a similar "EEGEI" motif with FLASH, a large protein involved in histone mRNA biogenesis[32,40]. FLASH[931–943] binds ARS2[147–871] with a $K_d$ of 0.5–5 µM depending on salt concentration[32], reminiscent of the Red1-Ars2 binding. Accordingly, in the human system, a K719A, K722, K734A triple mutant located at the EDGEI motif-binding surface of ARS2 (Supplementary Fig. 10a) abolished the hARS2-FLASH interaction[32].

A search for other human proteins containing possible ARS2-binding motifs with SlimSearch[41], identified several RNA biogenesis factors (Fig. 6f). These include three other Zn-finger proteins - ZC3H4, ZC3H6 and ZC3H18. Similarly to ZFC3H1, they contain two or three tandem copies of the motif. The strong effect of the single E23R mutation in one of the ZFC3H1 motifs (Supplementary Fig. 10b, lane 3) might indicate that individual motifs interact with different affinities depending on surrounding residues. NCBP3, linked to mRNA export[42] also possesses two separated copies of the motif. Notably, NCBP3 was shown to directly bind ARS2, and the C-terminal leg of ARS2 was

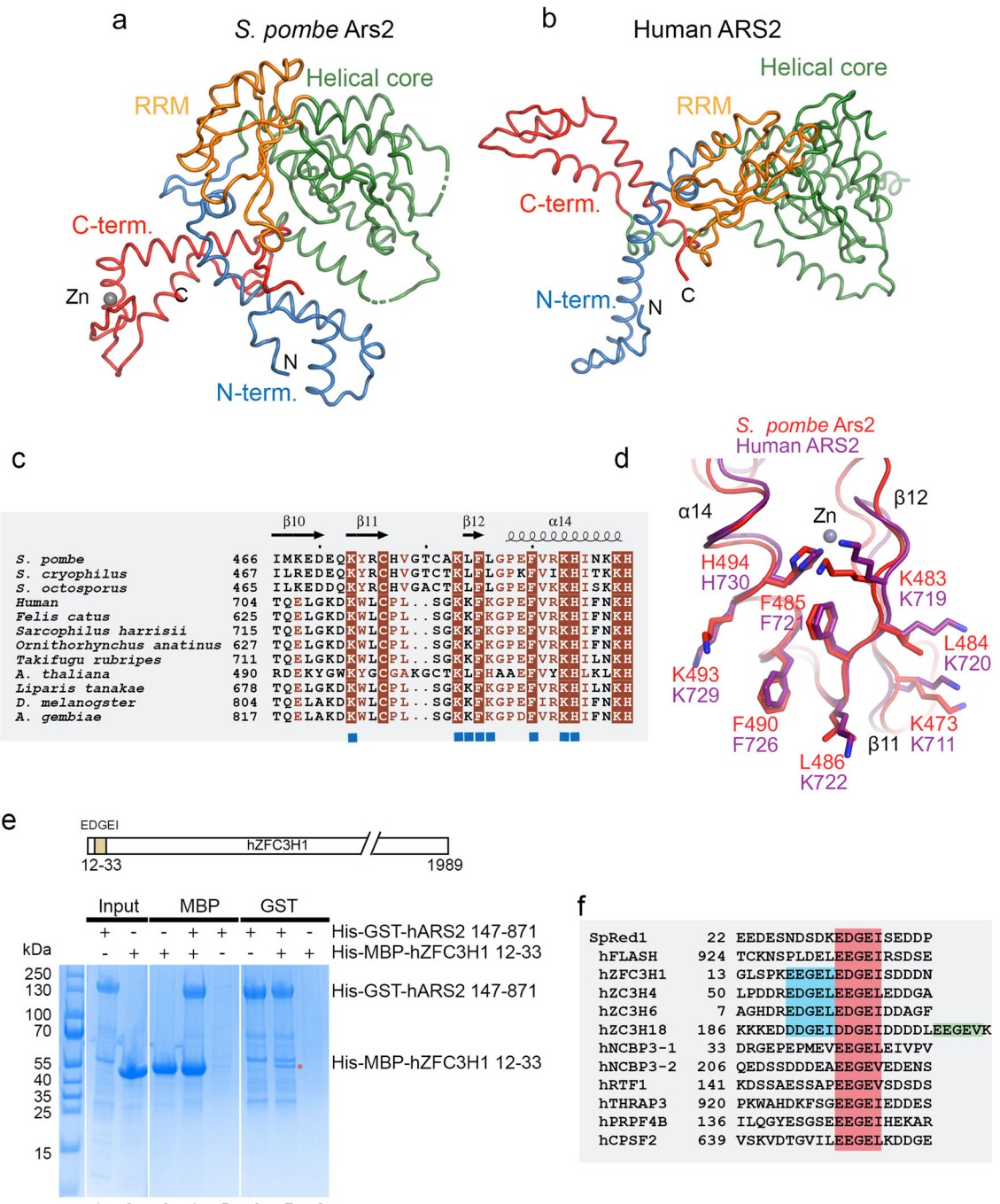

**Fig. 6 | The Ars2-Red1 interface is conserved in human. a** Comparison of the *S. pombe* and **b** human Ars2 structures. The proteins were superimposed using the helical core domain (in green) with r.ms.d.– 2.8 Å for 153 Cα atoms. **c** Sequence alignment of Ars2 proteins. Only the Zn-finger domain of the C-terminal leg is shown. Identical residues are in brown boxes. Blue squares indicate residues involved in the interaction with Red1. **d** The Red1 binding surface of *S. pombe* Ars2 is conserved in human ARS2. The secondary structure element labeling corresponds to *S. pombe* Ars2. **e** MBP- and GST-tag pull-down experiments with His-GST-tagged human ARS2$^{147–871}$ and His-MBP-tagged ZFC3H1$^{12–33}$ showing,

respectively, a mutual interaction (lanes 4 and 7). The proteins were first purified using Ni²⁺ resin (lanes 1, 2) and then co-purified on either Amylose or GST resins. The position of the EDGEI motif within hZFC3H1 is shown. Source data are provided as a Source Data file. **f** Sequence comparison of Red1 with human proteins containing possible ARS2-binding motifs. The ARS2 binding motif is highlighted in red. Additional copies of the motif are shown in blue and green. The SlimSearch[41] sequence analysis was done with a more relaxed sequence of the motif [ED][ED]GE[ILV], allowing also an aspartate in position 1 and a valine in position 5, likely compatible with the crystal structure of the Red1-Ars2 complex.

required for this interaction[32]. Finally, THRAP3, PRPF4B, RTF1, and CPSF2 contain one motif. In all these proteins, the conserved ARS2-binding motif is located in disordered regions as judged using Alpha-Fold2 predicted models. Importantly, ZFC3H1, ZC3H4, NCBP3, THRAP3, PRPF4B and CPSF2 were all found enriched in affinity purification/mass spectrometry (AP-MS) analysis with native hARS2

compared to the hARS2 K719A, K722A, K734A triple mutant in the ARS2 C-terminal leg[32].

Together, these results show that the *S. pombe* Ars2-Red1 interface is well conserved among species. From these results, we also hypothesize, that in addition to hZFC3H1 and FLASH, the ARS2 ED/EGEI/L binding surface is involved in connecting CBC-ARS2 bound-

RNAs to several other regulators of RNA metabolism, likely in a mutually exclusive way.

## Dimerization of Red1 C-terminus

To shed light onto the observed Red1 propensity to self-associate, we assessed which part of Red1 is required for its eventual oligomerization. In Y2H assays, the C-terminal Red1 construct (residues 390–712) interacted strongly with full-length Red1 (Fig. 7a). In addition, the C-terminal region was sufficient for the interaction when tested against the three Red1 fragments (Fig. 7a). We used AlphaFold2[35] to model the dimeric structure of the Red1 C-terminus. While most of the structure is predicted with low confidence, AlphaFold2 predicts an antiparallel dimeric coiled-coil corresponding to residues 470–524 with very high per-residue confidence score (pLDDT) and the predicted aligned error plot indicates high confidence of the mutual positions of the two protomers (Fig. 7b, Supplementary Fig. 11). Equivalent predictions were obtained for *S. octosporus* and *S. cryophilus* Red1 (Supplementary Fig. 11). The predicted structure revealed six complete heptad repeats engaged in classical coiled-coil packing (Fig. 7b–d). To test this prediction, a His-MBP-Red1[452–524] fusion protein was analyzed by size exclusion chromatography, where two elution peaks were observed (Supplementary Fig. 12a, b). MALLS revealed that the two peaks correspond to dimeric and monomeric Red1[452–524], respectively (Fig. 7e and Supplementary Fig. 12c). The dimeric Red1[452–524] remained dimeric upon reinjection on the gel filtration column (Fig. 7e and Supplementary Fig. 12d). Importantly, an L481R, I485R double mutation, based on the structure model (Fig. 7d), is able to abolish the Red1 dimerization (Supplementary Fig. 12e, f). Using Y2H we could show that the L481R, I485R double mutation essentially disrupted the dimerization of full-length Red1, when one or both protomers are mutated (Fig. 7f), but did not impact the binding of Ars2 nor Iss10 (Supplementary Fig. 13a).

We then investigated whether Red1 dimerizes in vivo. We constructed strains co-expressing the proteins Red1-Myc[13] and Red1-HA[3]. Co-IP experiments showed that indeed Red1 dimerizes within *S. pombe* (Fig. 7g). Co-expression of Red1-Myc[13] together with Red1-ΔCC-HA[3] (Δcoiled-coil corresponds to the deletion of residues 452–524) or Red1-L481R-I485R-HA[3] mutants showed that both mutations compromise Red1 dimerization but do not completely disrupt the interaction (Fig. 7g). We made other strains co-expressing Red1-HA[3] constructions together with Red1-TAP and obtained similar results (Supplementary Fig. 13b). We next explored whether the unique expression of either Red1 dimerization mutants resulted in Red1 loss of function. However, cell growth assays showed that *red1-ΔCC* or *red1-L481R-I485R* mutant cells growth is similar to wild-type cells (Supplementary Fig. 13c). In addition, examination of the RNA levels of transcripts known to be degraded by MTREC showed no significant changes of their level (Supplementary Fig. 13d). Thus, these results show that Red1 dimerizes in vivo and that the Red1 coiled-coil domain contributes to this dimerization in *S. pombe*.

Together, these results provide evidence for a coiled coil-mediated homo-dimerization of Red1. Given that Iss10, Ars2 and Mtl1 each interact with Red1 in a 1:1 or 2:2 stoichiometry, it is thus possible that these subunits are present in two copies within the MTREC complex (Fig. 8a). Importantly, AlphaFold2 also predicts, with high accuracy, coiled coil-based homodimers for the PAXT and NEXT complex scaffolding proteins hZFC3H1 and ZCCHC8, respectively (Supplementary Fig. 14). The dimeric architecture might thus be a conserved feature among all these RNA targeting complexes (Fig. 8b and Supplementary Fig. 15).

## Discussion

The *S. pombe* MTREC complex and its human counterpart PAXT target specific and aberrant nuclear transcripts for degradation by the nuclear RNA exosome[4,5]. While the core RNA Mtl1/MTR4 helicase has been extensively analyzed[15], the exact roles of the remaining subunits remain unclear. In this study, we describe atomic details of the Red1 subunit interactions with Iss10 of the Iss10/Mmi1 sub-module and Ars2 of the cap-binding sub-module (Cbc1, Cb2, and Ars2). In addition, we report a coiled-coil mediated homo-dimerization of Red1. Together with the recently reported crystal structure of the complex between Red1 and Mtl1[14], these results provide structural insights into the Red1 function as a scaffold of the MTREC complex. We also show that these MTREC features are conserved in the human analog PAXT complex.

Our NMR structure of the minimal Red1-Iss10 interaction complex revealed that Red1 uses a HTH motif to bind the Iss10 N-terminal helices. We identified a Red1 L205R mutant that disrupts the Red1-Iss10 interaction and analyzed its impact on MTREC functions in *S. pombe* cells. This mutation was sufficient to compromise the Red1-Iss10 binding in vivo, resulting in an Iss10 protein level decrease, indicating that the Red1 interaction is required for Iss10 stabilization. Accordingly, the amount of Iss10 was previously shown to be drastically reduced in *red1Δ* cells[26]. Noticeably, *red1-L205R* cells show a growth defect similar to *red1Δ* cells on minimal medium and at lower temperatures that is stronger than in *iss10Δ* cells, but the reason for this is currently unclear.

Red1 is crucial for nuclear exosome foci formation[23]. Specifically, this function has been attributed to region 196–245[23], which we now show forms a HTH domain required for Iss10 binding. Proper Red1 and exosome nuclear localization is also dependent on Iss10[6,23]. Accordingly, we show that the *red1-L205R* mutation affects both Iss10 and Red1 localization. The interaction of Iss10 with Red1 HTH domain is required for Iss10 recruitment to Red1 nuclear foci, may promote the clustering of Red1 nuclear foci and enable recruitment of the nuclear exosome. We note that the mislocalization of Red1 in *red1-L205R* mutant cells, which causes Red1 to localize in an increased number of foci within the nucleus, differs from Red1 mislocalization in *iss10Δ* cells, in which Red1 shows even distribution in the nucleus[6,26]. The reason for this difference is unclear but in *red1-L205R* cells the fact that Iss10 is still expressed may influence Red1 localization either due to residual direct interaction between Red1 and Iss10 or to the existence of an indirect interaction. In favor of the second possibility, affinity purifications of Red1 have suggested that Iss10-Mmi1 submodule may be connected to Red1 by several means[6–8].

Expression of the Red1-L205R mutant instead of WT Red1 leads to accumulation of meiotic mRNAs that is comparable to *iss10Δ* mutation. This phenotype is further strengthened when cells are grown in minimal medium, however, it still remains lower than in *red1Δ* cells. The Iss10-Red1 interaction is thus dispensable for most of the MTREC-based degradation of meiotic transcripts. Our Y2H assays also show that Iss10 interacts with Mmi1, known to directly bind to meiotic transcripts[21], indicating that Iss10 links Red1 to Mmi1 and hence to target transcripts. The same Y2H interaction was recently reported in ref. 14. In *iss10Δ* cells, the interaction between Red1 and Mmi1 is severely compromised but not completely lost[7,26]. It is thus possible that while Iss10-mediated interaction of Mmi1 with Red1 is important for MTREC activity, particularly in sub-optimal conditions, Mmi1 might also be recruited into MTREC by alternative interactions as proposed in ref. 14. The fact that the impact of the Iss10-Red1 interaction varies according to the environmental growth conditions might be linked to the Iss10 regulation by TORC1[27]. Further analysis of the Mmi1 interaction with MTREC and Iss10 regulation by TORC1 will help explain the mechanism of MTREC-mediated degradation of meiotic transcripts in *S. pombe*.

We also report the crystal structure of the complex between *S. pombe* Ars2 and a Red1 N-terminal peptide. The structure of Ars2 reveals four structured domains that resemble their counterparts in hARS2 but their mutual orientations are very different, highlighting considerable plasticity of the Ars2 architecture in different species. While ssRNA binding has been reported for the hARS2 RRM domain[32],

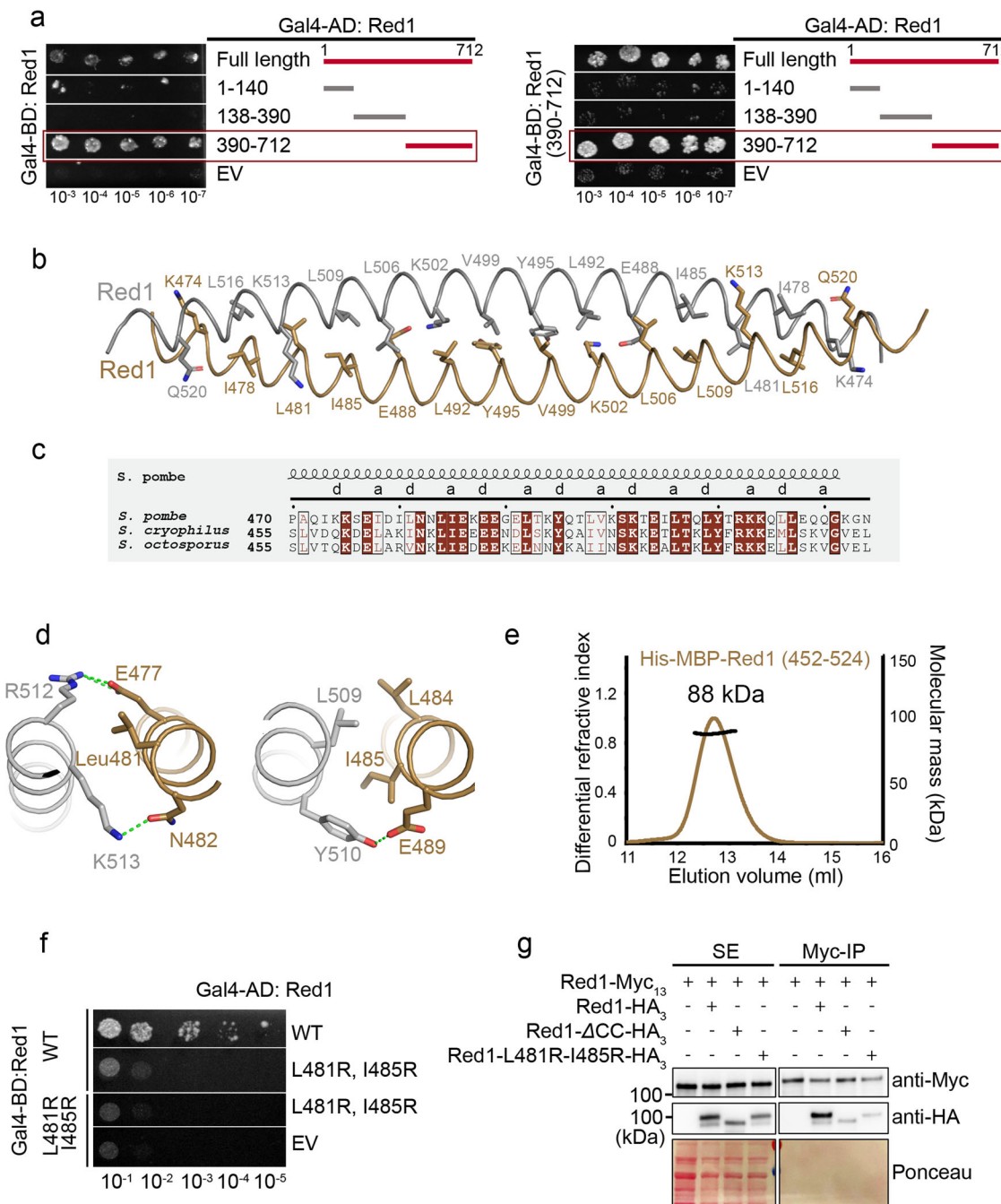

**Fig. 7 | Red1 homo-dimerization. a** Y2H mapping of the Red1 regions involved in its self-association. Growth was assayed on medium lacking histidine and with 30 and 50 mM 3AT. EV- empty vector. FL-Red1 (left panel) or Red1[390–712] (right panel) were used as a bait. **b** Ribbon representation of the Red1 dimeric coiled coil as modeled by AlphaFold2[35]. Residues in heptad positions indicated in **c** are shown as sticks. **c** Sequence alignment of Red1 proteins comparing different *Schizosaccharomyces* species. Only the sequence of the coiled-coil region is shown. Identical residues are in brown boxes, and conserved residues are highlighted in brown. Heptad positions are indicated. **d** Details of the Red1 dimeric coiled-coil interactions centered on L481 (right panel) and I485 (left panel). **e** Molecular mass determination of the His-MBP-Red1[452–524] by MALLS. The measured molecular mass of 88 kDa corresponds to a dimer. Calculated molecular mass of a monomer is 53 kDa. The sample was injected at 9 mg/ml. **f** Y2H assay showing that the L481R, I485R double mutant essentially disrupts the full-length Red1 dimerization when one or both protomers are mutated. Growth was assayed on medium lacking histidine and with 50 mM 3AT. EV- empty vector. **g** Co-immunoprecipitation experiments of Red1-Myc13 and Red1-HA3, Red1-ΔCC-HA3 or Red1-L481R-I485R-HA3, analyzed by western blot. Soluble extracts (SE) were prepared from exponentially growing cells in minimal liquid medium. Source data are provided as a Source Data file.

the *S. pombe* Ars2 RRM is tightly packed against the helical core, and its putative RNA binding residues are buried in the interface. hARS2 is involved in numerous interactions that are crucial for the determination of the target transcript fate[30]. Only the interaction of the flexible, extreme ARS2 C-terminus with CBC has so far been characterized in detail[43]. In the first structure of the Ars2 core with a

partner protein, we show how the Zn-finger domain of the Ars2 C-terminal leg interacts with the Red1 conserved EDGEI motif. We generated structure-based mutations in the Zn-finger domain and Red1 peptide that abolish the binding in vitro and showed that Red1 E32R mutation is sufficient to disrupt binding with Ars2 in vivo. The *Red1-E32R-TAP* mutant cells experience growth defect resembling the

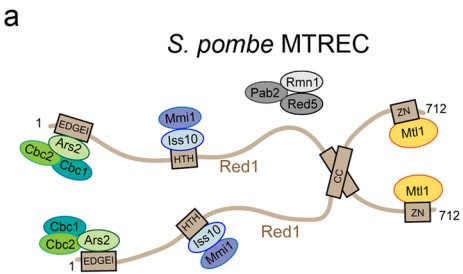

**Fig. 8 | MTREC/PAXT complex model. a, b** Schematic models of the MTREC and PAXT complexes, summarizing the structural and biochemical information obtained in this study and in ref. 14. The crystal structure between Mtr4 and Red1 was determined using *C. thermophilum* proteins and the key interacting residues are conserved in *S. pombe* Red1 and Mtl1 as well as in human MTR4 and ZFC3H1[14].

AlphaFold2[35] structure prediction of the human complex structure (Supplementary Fig. 15) indicates that that MTR4-binding region of ZFC3H1 might pack against a small helical domain. It is thus likely, that the MTR4-binding domain of ZFC3H1 covers residues 1140–1292. Similarly, in the AlphaFold2 prediction of Red1 (AF-Q9UTR8), this domain spans residues 575–703.

*red1Δ* cells, suggesting that Red1-Ars2 interaction is important for an optimal cellular growth. In addition, in mutant cells, Ars2 appears to be more diffused and less concentrated in nuclear dots. On the other hand, we did not observe any significant changes in the levels of several meiotic mRNAs and PROMPTs/CUTs known to be targets of MTREC, suggesting the interaction is not critical for the degradation of these targets by MTREC.

The Red1 EDGEI motif is conserved in human hZFC3H1[14], a subunit of the PAXT complex. We show that the structure of the C-terminal leg interacting surface is conserved between *S. pombe* and human ARS2. Correspondingly, we also show that hARS2 and hZFC3H1 interact, and the E23R mutation in hZFC3H1 (equivalent to E32R in Red1) is sufficient to disrupt the complex. This is consistent with the demonstration that the hARS2 construct corresponding to the C-terminal leg (628–876) is a major determinant in triggering RNA decay of endogenous PROMPTs transcripts and in tethering assays[44]. Indeed, in tethering assays, the ARS2 activity was reduced upon siRNA silencing of hZFC3H1, but not of ZC3H18 nor ZCCHC8[44]. Moreover, either a K719A, K722A, K734A triple mutation, or the single K719A mutation, in the C-terminal leg of ARS2 inhibited its RNA decay triggering activity[44]. Here, we show that residues K719 and K722 are directly involved in recognizing the Red1/ZFC3H1 EDGEI motif. It seems highly plausible that this activity of human ARS2 thus depends on its EDGEI motif-mediated interaction with hZFC3H1. In agreement, the same ARS2 triple mutation prevented interaction of hARS2 with PAXT and NEXT components in cell-based assays[32].

Human ARS2 has previously been shown to interact via its C-terminal leg with the EEGEI motif-containing FARB region of FLASH, a protein important for processing of replication-dependent histone mRNAs, and this interaction was abolished by the K719A, K722A, K734A triple mutation[32]. Our structure now provides an explanation for this interaction and implies a mutually exclusive binding of hARS2 to hZFC3H1/PAXT and FLASH. In addition to hZFC3H1 and FLASH, similar motifs are found in other important RNA regulators including ZC3H4, recently shown to restrict non-coding transcription[45], poorly characterized ZC3H6, the NEXT/PAXT subunit ZC3H18, NCBP3 connected to mRNA export[42] splicing factors THRAP3 and PRPF4B[46,47], RNA polymerase-associated protein RTF1, and the CPSF complex subunit CPSF2[48]. Interestingly, NCBP3[1–182] interacts with hARS2 in pull-down assays, which can be abrogated by the ARS2 C-terminal leg K719A, K722A, K734A triple mutation[32]. In addition, ZFC3H1, ZC3H4, NCBP3, THRAP3, PRP4B and CPSF2 all co-purify in AP-MS analysis with native hARS2 but not with the hARS2 K719A, K722A, K734A triple mutant[32]. While the exact nature of these interactions remains to be characterized, we speculate that the conserved binding surface on the ARS2 C-terminal leg accommodates multiple mutually exclusive interactions with competing partner proteins that exhibit the ARS2 binding motif 'EE/DGEI/L', and that this interaction is important for fate determination of targeted RNAs.

Finally, we also demonstrate the ability of Red1 to dimerize via its conserved coiled-coil region both in vitro and in *S. pombe*. Given that Iss10, Ars2, and Mtl1 can each bind to a single Red1, it is possible that the MTREC complex might exert its function as a dimeric assembly, where at least these subunits are present in pairs. This would be consistent with the reported dimerization of Mmi1[49], which is a direct binder of meiotic transcripts in *S. pombe*[20–22,50]. Mmi1 is dimerized via its interaction with its dimeric partner Erh1 and this is important for meiotic transcript degradation by MTREC[49]. Functional analyses of Red1 mutants that lost the capacity to homodimerize in vitro and in Y2H assays did not reveal any significant loss of function for the *S. pombe* cells expressing these Red1 mutants. Importantly, our co-immunoprecipitation experiments showed that in *S. pombe* these Red1 mutants can still partially dimerize, indicating that in vivo Red1 can dimerize by at least one other mechanism. Better understanding of the role of Red1 dimerization within MTREC will thus require further characterization. Interestingly, AlphaFold2 modeling predicts with high confidence dimeric coiled-coils also for hZFC3H1 of the PAXT and ZCCHC8 of the NEXT complex. Dimeric architectures might thus be a conserved feature of these RNA targeting complexes.

In conclusion, our structural and in vivo analyses of the MTREC/PAXT scaffolding proteins provide important insights towards elucidation of the molecular architecture and function of these essential RNA regulators.

## Methods

### Yeast two-hybrid (Y2H) Assay

Genes coding for MTREC subunits and their variants were cloned into pDONR221 vector using BP clonase recombination (Life Technologies). The inserts were then introduced into Y2H destination vectors (pDEST32 for bait and pDEST22 for prey) using Gateway cloning system (ProQuest-Life Technologies). Bait and prey plasmids were co-transformed into MaV203 yeast strain and plated on Nitrogen-base (NB) agar medium containing 2% glucose, Histidine, Adenine and Uracil and incubated for 48 h at 30 °C. Co-transformed colonies were replica-plated on the same medium and incubated for two more days at 30 °C. Co-transformants were then assayed for *HIS3* reporter gene activation, by plating them on NB agar medium (2% glucose, Ade, Ura) supplemented with different concentrations (5–50 mM) of 3-aminotriazole (3AT; Sigma-Aldrich) histidine biosynthesis inhibitor. Plates were incubated at 30 °C and growth was monitored. Control plasmids from ProQuest™ two-hybrid system (Life Technologies). Co-transformation with empty prey vector was used as negative control. In the initial screen, Krev/RalGDS interaction was used as a positive control, and RalGDS prey vector was also used as a mock control.

### Pull-down assay

All variants of Red1 were cloned as His-MBP fusions into the pETM41 vector. Iss10 constructs were cloned as N-terminal Strep-tag fusions

into pACYCDuet (Novagen). Combinations of Red1 and Iss10 constructs were co-expressed in *E. coli* BL21Star (DE3, Invitrogen). Following cell disruption, the Red1/Iss10 proteins-containing supernatants were loaded onto a Strep-Tactin XT resin (IBA) that was then extensively washed with a buffer containing 20 mM Tris pH 8, 100 mM NaCl and 5 mM β-mercaptoethanol. Bound proteins were eluted by the addition of 50 mM of D-Biotin, and analyzed on 15% SDS-PAGE. For analysis of the impact of Red1 mutants on the interaction with Iss10, half of each culture was used for Strep-tag pull-down assays, as described above, and the other half for MBP pull-down assays. Supernatants were applied on Amylose resin (NEB) that was extensively washed and the bound proteins were eluted with the addition on 10 mM maltose.

Ars2-Red1 complexes were analyzed in Strep-tag pull-down assays. Ars2[107–527] and its variants were cloned to contain an N-terminal His-MBP- and C-terminal Strep-tag into pETM41. His-MBP fusions of Red1 proteins and Ars2 variants were expressed individually in *E. coli* BL21Star (DE3) and purified using a Ni[2+] chelating Sepharose (GE Healthcare). Strep-Ars2 proteins were applied on Strep-Tactin XT columns and washed extensively. Equal amounts of Ni-purified His-MBP Red1 proteins were added. Columns were extensively washed with a buffer containing 20 mM Tris pH 8 and 100 mM NaCl and bound proteins were eluted with the addition of 50 mM of D-Biotin and analyzed by 12–15% SDS-PAGE.

For analysis of the interaction between human ARS2 and ZFC3H1, hARS2[147–871] was cloned into pETM30 as a His-GST fusion and variants of ZFC3H1 as His-MBP fusions into pETM41. The proteins were expressed individually in *E. coli* BL21Star (DE3) and purified on a Ni[2+] chelating Sepharose (GE Healthcare). His-GST-ARS2 was applied onto a GST resin (GE Healthcare) and His-MBP-ZFC3H1 onto an Amylose resin (NEB). After extensive washing, His-GST-ARS2 was added onto the ZFC3H1-bound MBP column, while His-MBP-ZFC3H1 was added onto the ARS2-bound GST resin. Columns were washed with a buffer containing 20 mM Tris pH 8 and 100 mM NaCl, the proteins were eluted by the addition of 10 mM reduced glutathione or 10 mM maltose, respectively, and analyzed on 15% SDS-PAGE.

### Isothermal titration calorimetry (ITC)
ITC experiments were performed at 25 °C using an ITC200 microcalorimeter (MicroCal). Experiments included one 0.5 µl injection and 15–20 injections of 1.5–2 µL of 0.4–0.8 mM His-MBP-Red1 (1–53, 20–40 or 20–40 E32R) or the Red1[20–40] peptide into the sample cell that contained 30–40 µM Ars2 (FL, 58–609 or 107–527) in 20 mM Tris pH 8.0 and 100–200 mM NaCl. The initial data point was deleted from the data sets. Binding isotherms were fitted with a one-site binding model by nonlinear regression using the Origin software, version 7.0 (MicroCal).

### Multi angle laser light scattering
Size exclusion chromatography (SEC)-Light scattering (LS) experiments were conducted at 4 °C on an HPLC chromatography system consisting of a degasser DGU-20AD, a LC-20AD pump, an autosampler SIL20-ACHT, a communication interface CBM-20A and a UV–Vis detector SPD-M20A (Schimadzu, Kyoto, Japan), a column oven XL-Therm (WynSep, Sainte Foy d'Aigrefeuille, France), a static light scattering miniDawn Treos, a dynamic light scattering DynaPro Nanostar and a refractive index Optilab rEX detectors (Wyatt, Santa-Barbara, USA). The analysis was carried out with the software ASTRA, v5.4.3.20 (Wyatt, Santa-Barbara, USA). Samples of 20 µL were injected at 0.5 mL min⁻¹ on a Superdex 200 10/300 GL (GE Heathcare), equilibrated with 20 mM Tris-HCl pH8, 100 mM NaCl, 5 mM β-mercaptoethanol. Bovine Serum Albumin, at 2 mg mL⁻¹, in PBS buffer was injected as a control.

### Protein Expression and Purification
His-MBP-Red1[192–236] and Strep-Iss10[1–45] were co-expressed in *E. coli* BL21Star (DE3) cells. After affinity chromatography using the Ni[2+] chelating resin (GE Healthcare) in a buffer containing 20 mM Tris pH 8, 200 mM NaCl and 5 mM β-mercaptoethanol, the His-MBP-tag on Red1 was cleaved off by TEV protease. The complex was further purified on Strep-Tactin XT resin and Superdex 200 size exclusion chromatography.

Ars2[68–183] was cloned into pProEXHTb (Invitrogen) as a His-tag fusion and Ars2[206–531] into pRSFduet vector (Novagen) as a Strep-tag fusion. The two fragments were co-expressed in *E. coli* BL21Star (DE3, Invitrogen) and the complex purified by affinity chromatography using the chelating Ni[2+] Sepharose (GE Healthcare). After His-tag cleavage with the TEV protease, the two fragments were co-purified on Strep-Tactin XT resin (IBA). The final purification step was a Superdex 200 size-exclusion chromatography using a buffer containing 20 mM Tris pH 8, 100 mM NaCl and 5 mM β-mercaptoethanol.

### NMR Spectroscopy
NMR spectra were recorded at 298 K using a Bruker Neo spectrometer at 700 MHz or 800 MHz, equipped with a standard triple resonance gradient probe or cryoprobe, respectively. Bruker TopSpin version 4.0 (Bruker BioSpin) was used to collect data. NMR data were processed with NMR Pipe/Draw[51] and analyzed with Sparky 3 (T.D. Goddard and D.G. Kneller, University of California).

### Chemical shift assignment
Backbone ¹Hᴺ, ¹Hα, ¹³Cα, ¹³Cβ, ¹³C′ and ¹⁵Nᴴ chemical shifts for the minimal Iss10-Red1 heterodimer were assigned from a sample of 500 µM [¹³C,¹⁵N]Red1[192–236]-Iss10[1–45] using 2D ¹H,¹⁵N-HSQC, 3D HNCO, 3D HNCACO, 3D HNCA, 3D HNCACB, 3D CBCACONH, 3D HNHA and 3D HACACONH spectra. Using the same sample, aliphatic side chain protons were assigned based on 3D H(C)(CO)NH-TOCSY (40 ms mixing time), and 3D (H)C(CO)NH-TOCSY (40 ms mixing time) spectra. This was complemented by 2D ¹H,¹³C-HSQC, 3D (H)CCH-TOCSY (16 ms mixing time), and 3D (H)CCH-TOCSY (16 ms mixing time) spectra collected on the sample after exchange to 99% (v/v) D₂O. Assignment of sidechain asparagine δ2 amides and glutamine ε2 amides used a 3D ¹⁵N-HSQC-NOESY (120 ms mixing time). Stereospecific ¹H-¹³C assignment of leucine and valine methyl groups used a 2D constant-time ¹³C-HSQC spectrum with a 150 µM sample of 10% ¹³C-labeled Red1[192–236]-Iss10[1–45] in 99% D₂O. Aromatic ¹H chemical shifts were assigned using the same sample with 2D TOCSY (60 ms mixing time) and 2D DQF-COSY spectra.

### NMR structure calculation
Structure ensembles were calculated using Aria 2.3/CNS1.2[52,53]. The ¹H distances were obtained using NOE crosspeaks from a sample of 500 µM [¹³C,¹⁵N]Red1[192–236]-Iss10[1–45] using a 3D ¹⁵N-HSQC-NOESY (120 ms mixing time) and 3D ¹³C-HSQC-NOESY (120 ms mixing time). Distance restraints were also obtained from the 150 µM sample of Red1[192–236]-Iss10[1–45] in 99 % D₂O from a 2D ¹H,¹H-NOESY (120 ms mixing time) spectrum. Protein dihedral angles were obtained by using TALOS-N[54] and SideR[55,56]. Hydrogen bond restraints (two per hydrogen bond) were introduced following an initial structural calculation and include only protected amides in the center of helices. Starting at iteration four in the structure calculations, residual dipolar coupling (RDC) values were included by measuring interleaved spin state-selective TROSY experiments. The aligned sample contained 17 mg/mL Pf1 phage (Asla Biotech). RDC-based intervector projection angle restraints used $D_a$ and $R$ values of 8 and 0.33, respectively. Final ensembles were refined in explicit water and consisted of the 20 lowest energy structures from a total of 100 calculated models. Complete refinement statistics are presented in Supplementary Table 1.

## X-ray structure determination

Pure Ars2[68–183,206–531] was concentrated to 8 mg ml$^{-1}$ in a buffer containing 20 mM Tris pH 8, 100 mM NaCl, and 5 mM β-mercaptoethanol using Amicon ULTRA concentrator (Millipore). The pure protein was supplemented with a three-fold molar excess of Red1 peptide (20-KNEEDESNDSDKEDGEISEDD-40) (PSL GMBH). The complex was crystallized using hanging drop vapor diffusion method at 5 °C. The best diffracting grew within three days in a solution containing 200 mM Ammonium Formate and 20% (w/v) PEG3350. For data collection at 100 K, crystals were snap-frozen in liquid nitrogen with solution containing mother liquor and 30% glycerol. Crystals of the *S. pombe* Ars2[68–183,206–531] - Red1[20–40] complex belong to the space group *P*2$_1$ with the unit cell dimensions *a* = 64.670, *b* = 128.281, *c* = 89.3 and *β* = 108°. The asymmetric unit contains a 2:2 Ars2/Red1 heterotetramer with a swapped C-terminus and has a solvent content of 63%. A complete native dataset was collected to a 3.1 Å resolution, partially extending to 2.8 Å, on the ESRF beamline ID30A-3 using the MXcuBE3 software (ESRF). The data were processed using autoPROC[57]. Phases were obtained by molecular replacement using PHASER[58] with an adjusted AlphaFold2 model of *S. pombe* Ars2 (AF-094326) as a search model. The initial map was improved using the prime-and-switch density modification option of RESOLVE[59]. After manual model rebuilding with COOT[60], the structure was refined using Refmac5[61] with NCS restraints to a final $R_{free}$ of 30.2% and an $R_{work}$ of 24.5% (Supplementary Table 2) with 99.8% residues in allowed (94% in favored) regions of the Ramachandran plot, as analyzed by MOLPROBITY[62]. A representative part of the $2F_o - F_c$ and omit $F_o - F_c$ electron density omit maps calculated using the refined model with and without the Red1 peptide, respectively, are shown in Supplementary Fig. 7b.

## Standard yeast genetics

Strains were constructed by standard genetic techniques (Moreno et al., 1991), cultured at 30 °C and under agitation at 180 rpm, in YEA (Rich medium) or Edinburgh minimal medium with appropriate supplements (Minimal medium). Genotypes of strains used in this study are listed in Supplementary Table 3.

Red1-TAP, Ars2-GFP and Iss10-GFP *S. pombe* cells were obtained using the PCR-based gene targeting method[63]. Positive transformants were selected by growth on YEA medium containing the appropriate antibiotic and confirmed by genomic PCR. Cells expressing point mutants of Red1 protein were generated in two steps from parental cells expressing *red1-TAP*. First, *ura4* gene was integrated into *red1* locus in order to generate *red1-Δ30–207*. Positive transformants were selected by growth on medium deprived of uracil and validated by genomic PCR. Then, *red1Δ 30–207::ura4 +* cells were transformed with synthesized DNA fragments of *red1* (GeneArt) containing either the E32R or L205R mutations. Positive transformants were selected by growth on YEA-5FOA and validated by genomic PCR and DNA sequencing. Oligos used for the different strain generation are listed in Supplementary Table 4.

Cells co-expressing one of the different versions of Red1-TAP (WT, E32R or L205R) together with Ars2-GFP or Iss10-GFP were obtained by crossing cells with appropriate genotypes using random spore analysis and selection on the appropriate media. *S. pombe red1-TAP iss10Δ* cells were obtained by crossing *iss10Δ* strain with *red1-TAP* strain and growth on the appropriate selective medium after random spore analysis.

To obtain plasmids expressing the Red1 mutants, the pJRXL-Pnmt81 plasmid expressing *red1-HA* (WT) was subjected for mutagenesis using Q5$^R$ Site-Direct mutagenesis Kit (NEB) using the oligo GTAGCTGCGGTTGACAACAA to generate *Pnmt81-red1Δ452–524* plasmid, and the oligo GAGATTGACATTAGAAATAACTTAAGA to generate *Pnmt81-red1-L481R-I485R* plasmid.

## Cell growth spotting assay

Spotting assays were done from 10$^7$ cells of exponentially growing cultures (OD$_{600nm}$ < 2) in minimal media. A 1/10th serial dilution of cells was dropped on rich or minimal media and grown at different temperatures (30, 25, and 18 °C). Pictures of the plates were acquired (Syngene bioimaging camera) after 3 days for the plate grown at 30 °C, 5 days for the plates grown at 25 °C and 10 days of grown at 18 °C.

## Live cell imaging

Red1-TAP (WT), Red1-TAP-E32R (E32R) and Red1-TAP-L205R (L205R) *S. pombe* cells co-expressing Ars2-GFP or Iss10-GFP were grown on liquid minimal medium. Live microscopy analysis was performed when cells reached an OD$_{600}$ of 0.5–0.8. Images were acquired with a Zeiss Apotome microscope (Carl Zeiss MicroImaging) and using a 63× oil immersion objective with a numerical aperture of 1.4 (WD 190, DICIII and GFP filters). Raw images were analyzed using the AxioVision software (Carl Zeiss MicroImaging), and processed using ImageJ.

## Immunofluorescent microscopy

The same strains used for the live cell imaging were used to assess the Red1-TAP localization by detecting its TAP tag using immuno-fluorescence experiments. Briefly, 20 ml of *S. pombe* cell cultures in logarithmic phase (OD$_{600}$ = 0.5–0.8) and grown in minimal medium were fixed with 3.8% paraformaldehyde at room temperature for 30 min, then washed with 10 ml PEM (100 mM Pipes, 1 mM EGTA, 1 mM MgSO$_4$ pH6.9). The cells were then transferred to 1.5 ml Eppendorf tubes with 1 ml of PEM and washed 2 more times. Cells were resuspended in 1 ml PEMS (PEM, 1 M D-Sorbitol) with 0.25 mg Novozym (Sigma) and 0.25 mg Zymolyase (Sigma) and incubated at RT (ideal @30 °C) until >90% cells are digested, then washed 3 times in 1 ml PEMS and spin 30 s at 500 g. Next, the cells were permeabilized 2 min with PEMT (PEM 1% Triton X-100), washed once with 1 ml of PEM and then blocked with PEMBAL (PEM, 1% BSA, 100mM L-Lysine) for 30 min. Detection of Red1-TAP was obtained by incubating the cells for 2 h at RT with the anti-TAP primary antibody (Thermo scientific #CAB1001) (1:100) in PEMBAL on a rotating wheel, the cells were washed 3 times 10 min with PEMBAL. The cells were then incubated over night at 4 °C on rotating wheel (or 2 h at RT) with a secondary antibody coupled to a fluorescent dye (DyLight® 549 (VECTOR lab # DI 1549) (1:400) dilution in PEMBAL, cells were washed 3 times 10 min with 1 ml of PEMBAL then resuspended in 100 μl PEMBAL and mounted on poly L-lysine coated coverslips. Fluorescence microscopy imaging were done with a Zeiss Apotome microscope (Carl Zeiss MicroImaging) and using a 63× oil immersion objective with a numerical aperture of 1.4 (WD 190, DICIII and GFP filters). Raw images were acquired using the AxioVision software (Carl Zeiss MicroImaging), and processed using ImageJ.

## Protein Co-immunoprecipitation

For the Red1-TAP/Iss10-GFP and Red1-TAP-Ars2 GFP CoIPs, 20 ml of *S. pombe* cells were grown to mid-log phase in minimal medium at 30 °C, harvested, and flash-frozen in liquid nitrogen prior to protein extract preparations. Extracts prepared from cells expressing epitope-tagged proteins under the control of native gene promoters. Red1-TAP WT or mutated versions (E32R, L205R) were used for immunoprecipitations to assess the interaction between Red1-TAP (WT or E32R point mutant) with Ars2-GFP, and between Red1-TAP (WT and L205R point mutant) and Iss10-GFP. Total protein extracts were prepared from cells lysated mechanically using glass beads (2× 60 sec) in IP-Lysis Buffer (50 mM HEPES-NaOH pH = 7.5, 5 mM MgCl$_2$, 20 mM β-glycerophosphate, 10% glycerol, 1 mM EDTA, 1 mM EGTA, 50 mM NaF, 0.1 mM Na$_3$Vo$_4$, 0.2% NP-40, 150 mM NaCl, 1 μg LABP, 1 mM Benzamidine, 1 mM PMSF). Soluble extract cleared of cellular debris was incubated with IgG sepharose (GE Healthcare) for 2 h at 4 °C on a rotating wheel. Afterwards, beads were washed three times with the Lysis Buffer. Protein elution was performed using 2× Laemmli buffer (50 mM Tris-HCl, pH

6.8, 10% ß-mercaptoethanol, 1% SDS, 10% glycerol and 0.1% bromophenol blue) and by heating the samples 10 min at 70 °C. The IP samples and 5% of the input were then separated by SDS-PAGE, transferred to nitrocellulose membranes and analyzed using appropriate primary (anti-TAP -Thermo scientific #CAB 1001, anti-GFP -Sigma # 11814460001) and secondary antibodies (HRP goat anti-Mousse -Dako #P0447, HRP goat anti-Rabbit -Dako #0448). Antibodies were diluted in TBS (0.1% Tween20- 1% fat dry milk; 1:1000 for the primary and 1:10,000 for the secondary), and membranes were washed in TBS- 0.1% Tween20.

For Iss10-GFP/Red1-CBP CoIPs, extracts were prepared from cells expressing the epitope tagged proteins under the control of native gene promoters at their endogenous locus and expressing Red1-TAP Red1-L205R-TAP. Whole cell extracts were prepared as described above with exception that the cell lysates were incubated at 4 °C 1 h on a rotating wheel with 20 U/ml of Arctic-TEV protease (2279 Protean) to remove the two ProteinA tag and then incubated with 20 µL of GFP-Trap$^R$ magnetic beads (gtd-20 Chromotek) for another hour at 4 °C on a rotating wheel. For Red1-Myc/Red1-HA CoIPs, extracts prepared from cells expressing Red1-Myc or Red1-TAP epitope tagged proteins *red1* at endogenous locus, and WT, *red1Δ452–524* or *red1-L481R-I485R* Red1-HA proteins from a plasmid under pnmt-81 promoter.

Red1-TAP/Red1-HA CoIPs were done as described above. Red1-Myc/Red1-HA CoIPs were done as described above except for the whole cell protein extracts that were incubated with 10 µg of mouse anti-Myc 9E10 (Covance) at 4 °C 1 h on a rotating wheel before incubation with 10 µL of Dynabeads™ Protein-G magnetic beads (Thermofisher) for another hour at 4 °C on a rotating wheel. The washes, elution and detection are done as described above using rabbit anti-HA (ab9110 Abcam) and anti-Myc for the detection of Red1-HA and Red1-Myc, respectively.

Enhanced chemiluminescence (ECL) detection was performed using reagents from Biorad (Clarity Western ECL kit) and revealed using ChemiDocMPSystem (BioRad) or the Fusion FX (Vilber) camera. Signals were quantified with the corresponding software, the image-Lab software (Biorad) or the Fusion-Capt software (Vilber Lourmat).

## Quantitative RT-PCR

Total RNA was extracted from *S. pombe* cells using hot phenol-acid. Briefly, 25 ml of logarithmic phase ($OD_{600}$ = 0.5−0.8) cells cultivated in minimal medium were collected by centrifugation 5 min at $1500 \times g$. Cells were washed with 1 ml of $dH_2O$, then lysed by the addition 750 µl of TES (10 mM Tris pH 7.5; 10 mM EDTA pH 8; 0.5% SDS) and 750 µl cold acidic phenol-chloroform (Sigma P-1944), incubated at 65 °C for 15 min, and vortexed 10 s every 3 min during the heat incubation. Samples were then placed on ice for 1 min, vortexed 20 s and centrifuged for 15 min at $12,000 \times g$ at 4 °C. A total of 700 µl of the upper phase was recovered into a phase-lock (heavy) tubes (Eppendorf) and a second phenolic treatment was performed by the addition of 700 µl of acidic phenol-chloroform and mixing thoroughly by inverting the tubes. Samples were then centrifuged 5 min at $12,000 \times g$ at 4 °C. The upper phase was transferred to new phase-lock tubes, and 700 µl of chloroform (Sigma) was added to the samples, mixed thoroughly again by inverting the tubes, and centrifuged 5 min at 12,000 g at 4 °C. The aqueous phase (500 µl) was carefully transferred to a 2 ml Eppendorf tubes containing 1.5 ml of 100% cold EtOH (−20 °C) and 50 µl of 3 M NaAc pH5.2, the mix was vortexed and incubated 15 min at −80 °C then centrifuged for 10 min at $12,000 \times g$, 4 °C. The pellet was washed in 75% (v/v) EtOH and centrifuged at $12,000 \times g$ for 5 min, 4 °C. The pellet was then air-dried, dissolved in 100 µl RNase-free water, incubated at 60 °C for 10 min and stored at −20 °C. RNA concentration and the A260/280 ration were determined using a NanoDrop ND-1000 UV spectrophotometer. RNA integrity was evaluated by electrophoresis on 1 % agarose gel using the GelGreen Nucleic Acid Stain (Biotium) dye following the manufacturer's instructions.

Reverse transcription and PCR were performed on 2 µg of purified RNA using the Transcriptor Reverse Transcriptase (Roche). RNA was first subjected to DNAse (Roche) treatment in the presence of 5X DNAse buffer (100 mM Tris pH 8, 10 mM $MgCl_2$), 100 mM DTT (Invitrogen) and RNase inhibitor (Thermo Scientific) for 20 min at 37 °C. Next 1.9 µl of 50 mM EDTA were added to the RNA-DNAse mixture and incubated 10 min at 70 °C. RNA was then hybridized with 50 µM of random hexamer primer (Invitrogen) in the presence of 50 mM $MgCl_2$ for 10 min at 65 °C. The cDNA synthesis was performed by adding 10 mM dNTP (Thermo Scientific), RNase Inhibitor (Thermo Scientific), RT buffer (Roche) and Reverse Transcriptase (Roche) and subsequent incubation for 10 min at 25 °C, 40 min at 55 °C and 5 min at 85 °C, according to the manufacturer guidelines.

qPCR was performed on a LightCycler480 machine (Roche). 4 µl of the diluted (1:4) cDNAs were used with 0.1 µM of each primer (forward and reverse) and 2X MESA Blue qPCR mix (Eurogentec) in a 20 µl final qPCR reaction. DNA amplification was done using the following program: 15 min incubation at 95 °C, followed by 40 cycles of 95 °C for 15 s, 60 °C for 15 s and 72 °C for 15 s. Oligos used for the qRT- PCRs are listed in Supplementary Table 4.

## Statistics and reproducibility

The pull-down experiments in Figs. 2i, 6e and Supplementary Figs. 1e, 7d, 10b, c were performed twice. The pull-down experiment in Fig. 5d was performed three times. Gel filtration experiments in Supplementary Fig. 2h, 12b, f were done twice. The gel filtration experiments in Supplementary Fig 4b–e and Supplementary Fig. 5d were done at least three times. The co-immunoprecipitation experiments in Fig. 3a, Supplementary Fig. 3a, Fig. 5g, Fig. 7g and Supplementary Fig. 13b were performed two times. The RT-qPCR experiments in Fig. 3f, Supplementary Figs. 3d, 8b, c, and 13d were performed three times. The RT-PCRs experiments in Supplementary Fig. 8d were performed three times. The immunofluorescence experiments in Figs. 3c, d, 5i, and Supplementary Fig. 8f were performed three times. The spotting assays in Fig. 3b, Supplementary Fig. 3b, c, Fig. 5h, Supplementary Figs. 8a and 13c were performed three times.

## Reporting summary

Further information on research design is available in the Nature Research Reporting Summary linked to this article.

# Data availability

The data that support this study are available from the corresponding author upon request. The structure ensemble of the Red1$^{192–236}$-Iss10$^{1–45}$ heterodimer generated in this study has been deposited at the Protein Data Bank (http://www.ebi.ac.uk/pdbe/) with accession ID 7QUU. Corresponding chemical shift assignments have been deposited in the Biological Magnetic Resonance Data Bank (http://bmrb.wisc.edu/) under BMRB accession number 34702. The atomic coordinates and structure factors of the *S. pombe* Ars2-Red1 complex determined in this study have been deposited under the PDB accession codes 7QY5. Source data are provided with this paper.

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

## Acknowledgements

J.K. was funded by the CNRS ATIP-Avenir program. A.V., S.C., and J.K. were funded by ANR MTREC (ANR-21-CE11-0021) and ANR RNAGermSilence (ANR-13-BSV2-0012) to A.V. Ariadna B. Juarez-Martinez was supported by the Labex GRAL (Grenoble Alliance for Integrated Structural Cell Biology) (ANR-10-LABX-49-01) and the People Program (Marie Curie Actions) of the European Union's Seventh Framework Program (FP7/2007–2013) under REA grant agreement PCOFUND-GA-2013-609102, through the PRESTIGE program coordinated by Campus France. Financial support from the Center National de la Recherche Scientifique (IR-RMN-THC Fr3050) is gratefully acknowledged. IBS acknowledges integration into the Interdisciplinary Research Institute of Grenoble (IRIG, CEA). This work used the platforms of the Grenoble Instruct-ERIC centre (ISBG; UAR 3518 CNRS-CEA-UGA-EMBL) within the Grenoble Partnership for Structural Biology (PSB), supported by FRISBI (ANR-10-INBS-0005-02) and GRAL, financed within the University Grenoble Alpes graduate school (Ecoles Universitaires de Recherche) CBH-EUR-GS (ANR-17-EURE-0003). We thank Caroline Mas for assistance and access to the biophysics platform, Luca Signor for mass spectrometry analysis, Aline Le Roy for assistance and access to the Protein Analysis On Line (PAOL) platform, the staff of the ESRF-EMBL (European Synchrotron Radiation Facility-European Molecular Biology Laboratory) Joint Structural Biology Group, particularly Andrew McCarthy and Matthew Bowler, for access to and help with the ESRF beamlines, Mylène Pezet from the IAB microscopy platform (MicroCell) for help with microscopy analyses as well as Daniel Perazza for the boxplot, statistical analyses and helpful comments. We thank the EMBL high-throughput crystallization facility (HTX). We thank Axelle Grélard, Estelle Morvan, and the structural biology platform at the Institut Européen de Chimie et Biologie (UMS 3033) for access to NMR spectrometers, equipment, and technical assistance. We thank Rachel Fitzgerald for help with initial validation of Red1-Ars2 interaction and Wiebke Schulze for providing hARS2 expression plasmids. We thank Ekaterina Flin for her help with statistical analyses.

## Author contributions

A.-E.F., S.A., and M.S. performed yeast two-hybrid experiments. A.-E.F., A.J.-M., and H.L. did all the in-vitro characterization. L.T.-T., A.R., and C.K. performed all the in-vivo experiments. H.L., J.K., and C.M. determined the NMR structure of Red1-Iss10 complex. A.-E.F. and J.K. determined the crystal structure of Red1-Ars2 complex. S.C. provided ARS2 expertize. M.S., S.C., C.M., A.V., and J.K. wrote the manuscript.

## Competing interests

The authors declare no competing interests.
