## [Peer Review File · Nature Communications]

Structural analysis of Red1 as a conserved scaffold of the RNA-targeting MTREC/PAXT complexReviewers' Comments:

Reviewer #1:

Remarks to the Author:

In the fission yeast *S. pombe*, the MTREC complex plays a crucial role for targeting of transcripts including meiotic mRNAs and cryptic unstable transcripts. MTREC is considered to be a counterpart of the PAXT complex in human, and its core complex is composed of a zinc-finger protein Red1 and an RNA helicase Mtl1. In this manuscript, Foucher and colleagues investigate the interaction mode of Red1 with MTREC submodules. They solve the structure of the Red1-Iss10 complex, which is involved in the silencing of meiotic transcripts, by NMR. The crystal structure of the Red1-Ars1 complex is also determined. Red1 interacts with Ars2 through the "EDGEI" motif, and this interaction is conserved between ZFC3H1, a human counterpart of Red1 and hArs2.

Overall, the work is sound and well written. The experiments presented are convincing and well conducted. This study advances our understanding of the target recognition mechanism of RNA-targeting complexes not only in yeast but also in higher eukaryotes including human. I have a few minor comments listed below.

Specific comments:

lines 124-125: The authors state: "Red1, together with the Mtl1 helicase, has been proposed to form the MTREC core, but the molecular details of its interactions with the MTREC sub-modules remain unknown". However, a similar line of experiments has been reported by Dobrev et al. (doi: 10.1038/s41467-021-23565-3). The authors should compare the results with those of the previous study.

Figure 3a: Although Iss10 is unstable in red1-L205R mutant cells, it is detectable to some extent in total cell extracts. Is it possible to immunoprecipitate Iss10-GFP and to examine the interaction between Iss10 and Red1-L205R?

Figure 3c: Red1-L205R localization differs from wild-type Red1 localization in the iss10 deletion background, in which Red1 shows even distribution in the nucleus (Egan et al. doi: 10.1261/rna.044479.114, Yamashita et al. doi:10.1093/nar/gkt763)

It has been demonstrated that Ars2 regulates several genes by targeting upstream long non-coding RNAs. Is expression of such genes (e.g., *pho1* and *byr2*) affected by the red1-E32R mutation? The significance of Red1 dimerization seems important, but remains ambiguous. How does non-dimeric L481R, I485R Red1 behave? Does it properly interact with Iss10 or Ars2? Does this mutation give an impact on the in vivo function of Red1?

Minor comments:

line 187: F215R should be Phe215.

Figure 2j: Gal4-AD (or BD?) should be Iss10.

Figure 5f: Gal4-AD (or BD?) should be Ars2.

line 337; *A. thaliana* SERRATE should be explained here.

Reviewer #2:

Remarks to the Author:

In their manuscript "Structural analysis of Red1 as a conserved scaffold of the RNA-targeting MTREC/PAXT complex" Foucher and co-workers expand the structural and functional knowledge about the interaction network of the *S. pombe* MTREC complex and its human equivalent PAXT.

While the core exosome and the subcomplexes TRAMP / NEXT have been comparably well studied, information about the human PAXT complex and the functionally homologous *S. pombe* MTREC complex is still incomplete with respect to their inner workings and functionality. This work focuses on the interactions of the central PAXT scaffolding protein Red1 with Iss10 and Ars2. Initially, the interaction network of Red1 within the MTRC complex is explored by Y2H assays partially complementing an earlier study. The authors provide first structural insight into the Red1-Iss10 interface by NMR. Also, they extend their prior structural study on human Ars2 now with the structure of *S. pombe* Ars2 in complex with Red1. The structural data now allows detailed comparison between the yeast and human complexes and rationalizes mutually exclusive interactions of hARS2 with various "ED/EGEI/L" motif-possessing RNA regulators such as hZFC3H1, hFLASH or hNCBP3. Lastly, this study sheds light on a potential dimerization of the MTREC/PAXT complexes via the respective Red1 and hZFC3H1 proteins. It is noteworthy that this study nicely implements AlphaFold. In summary, structural and in vivo data on the RNA-targeting MTREC/PAXT complexes presented in this work provides further insight into the respective complex architecture and function.

The work is original, well and comprehensively written, and parts of it are of high interest. The experimental work is technically sound and provides strong evidence for the conclusions made. The authors provide further details on the interaction map of the *S. pombe* MTREC complex. However, a large fraction of these interactions has been reported before, several claims for novelty of the data are not justified. However, the structures as well as the wealth of in vivo data are novel and make a valuable contribution to the field. This work will seed and alleviate follow up structural studies of the MTREC complex and partially, this work may be transferred to the related human PAXT complex.

In summary, I recommend a major revision of this article as a prerequisite for publication in Nature Communications. In particular, by the omission of published data by Dobrev et al., some results of this study are presented as novel which is not justified. An adequate citation of Dobrev et al. in the respective parts of this study and comparison has to be included to guarantee an adherence to good scientific practice.

Major points

Most of the Y2H interactions, e.g. Ars2-Red1 or Iss10-Red1 described on page 4, lines 125-135, have been already reported by Dobrev et al. (Nat. Commun 2021), which is cited as ref 14. This needs to be pointed out clearly and not sold as new data. Also, the role of Red1 as MTREC scaffolding protein has been proposed by Dobrev et al..

Most of the Y2H interactions, e.g. between Red1 and Iss10 were already narrowed down by Dobrev et al. (Nat. Commun. 2021). Again, this needs to be pointed out clearly and not sold as new data. I see it very critical not to mention the study by Dobrev et al. here since this is out already for a while. The novelty advertised here is simply not present.

Page 6, line 199: Quoting "While red1Δ cells showed only a moderate growth defect on solid rich medium at 30°C...". The prior characterization of this mutant by Dobrev et al. is omitted here. It has to be included and compared with this study.

Page 7, lines 217-231: Again, the authors sell their findings on the interface between Red1 and the short Ars2 motif as novelty. Although corroborated by gel filtration and ITC in this study, this interface was already published by Dobrev et al.

Figure 4c was already shown by Dobrev et al. in a similar fashion. This is not indicated. Likewise, the observation that the N-terminal motif of Ars2 in yeast and human is similar was already made in the earlier study.

Page 8, lines 275 – 281: I suggest to complement these findings with a PISA server analysis of the

interface (or parts of it) to maybe obtain more information about physiological relevance.

I would recommend to assess the effects of Red1 dimerization also in vivo based on the L481R, I485R mutants where dimer formation is abolished in vitro. It would be interesting to see a functional relevance or rule that out.

Minor points

The abstract is somehow missing a rational why this study was conducted.

Page 4, line 114 and Page 4, line 118: Rather use present, "demonstrate" instead of "demonstrated"

In Figure 2, f - h, the side chains of Iss10 should be coloured differently to avoid confusion with the Red1 backbone.

In Figure 2 j, the fusion partner of the GAL-4-BD construct is missing.

Page 6, line 179: "To this end, we mutated Ala198..." does not provide information to which residue it was mutated. The same for both hydrophobic residues. What is the rational behind these mutations?

Page 6, line 184: The authors may consider using one-letter-code for amino acids instead of three-letter-codes, however, not both. Figures come along less crowded with one-letter-coded amino acids.

Page 6, line 186: replace "purify on" by "purify employing"

Page 6, line 187: "using an Y2H assay"

Page 6, lines 188-190: Quoting "Ultimately, the L205R mutation was selected for in vivo studies (see below) and the integrity of mutated Red1192-236 was verified by gel filtration, where its elution profile was similar to the WT protein (data not shown)". I would not really call it "verification" without showing the WT gel filtration data.

Page 6, line 194: Replace "In order to evaluate in vivo the function of Red1-Iss10 binding interface" by "In order to evaluate in vivo the function of the Red1-Iss10 binding interface"

Page 7, line 214: rather "the Red1-Iss10 binding interface"

In figure 3, c,d, panels have different sizes. Figure size has to be increased - at the present panel size, contents are not comprehensively visualized.

In figure 3 f, error bars have to be coloured differently and should be in front of the bars for visibility.

Page 9, lines 297 - 298: Quoting: "The integrity of the Ars2 mutants was verified by gel filtration, with an elution profile similar to the WT protein (data not shown)." I recommend to include WT gel filtration data.

In Figure 5d, lane 6, the wt seems to bind sub-stoichiometrically? Is there an explanation?

Page 9, line 313: Rather "impact of the red1-E32R point mutation"

Page 9, line 323: Rather "As MTREC degrades a specific population"

Page 10, lines 326-329: "More diffused dots" cannot be reasoned from Figure 5i in my opinion.

Page 10, line 337: *A. thaliana* SERRATE has to be introduced to the readers first before comparing it with Ars2.

Page 13. Lines 462-463: This has also been implied by Dobrev et al.

The discussion may benefit from an overview figure that goes maybe even beyond fig 7f wr to complexity, e.g. extending it towards a hypothetical PAXT assembly.

Reviewer #3:

Remarks to the Author:

In their manuscript the authors mapped interactions between components of the MTREC complex and solved structure of the two peptides derived from Red1 and Iss10 as well as structure of fragments of Red1 and Ars2.

The data is of decent quality and the findings are interesting. However, I am appalled by the fact that most of the presented data has been published in Nat. Communications 9 months ago. The authors mention this manuscript, but (intentionally) fail to tell the reader that their findings are not novel. It can very well be that the authors got badly scooped, but to plainly pretend that the data in the literature is not there is not acceptable, especially since the time between the ref 14 paper and today is significant (~9 months). As such, I cannot recommend the current manuscript for publication. I suggest that the authors remove the known parts and write a manuscript for submission in a more specialized journal that focus on the novel structural data and the functional assays only.

The clear neglecting of the data in Ref 14 is clear from e.g. the following statements:

"The subunit composition of the *S. pombe* MTREC and human PAXT complexes has been characterized by proteomics". This sentence discredits previous work by the Sinning and Fischer labs (ref 14) that have made use of Y2H and Y3H approaches and those labs revealed a detailed analysis of MTREC submodule interactions.

"In MTREC, the Red1 protein was suggested to form a scaffold of the complex, interacting with multiple subunits and linking individual sub-modules together^{6,7,14}". These previous data do not only suggest something, the unambiguously show an important role of Red1.

". Red1 interacts with the Mtl1 Arch domain via its zinc-finger containing domain at its C-terminus¹⁴". This has been mapped in ref 14 accurately to Red1 (Ars2, Iss10, Pla1, Rmn1).

"How the Red1-Mtl1 core interacts with the individual MTREC sub-modules and what is their function in MTREC/PAXT activity, however, remains unclear", data is in ref 14 paper.

"How the CBC-ARS2 complex is integrated is currently unknown. ". A clear motif has been identified in Ref. 14.

". We mapped minimal Red1 regions required for its interaction with Iss10 and Ars2, respectively." This was done before in Ref 14.

In my opinion also many paragraphs in the results section do not provide new data or insights, inducing: "Red1 forms multiple interactions within the MTREC complex ", "Red1 interacts with the N-terminus of Iss10 ", "Red1 interacts with Ars2 via its conserved N-terminal motif" and many parts of the other sections.

The structure of the complex of the Red1 and Iss10 peptide is novel, but the interacting residues had

been determined before. The structure of the Ars2 and a Red1 N-terminal peptide appears novel. The PDB validation report, however, clearly state "This wwPDB validation report is NOT for manuscript review".

Nature Communications manuscript NCOMMS-22-01923

Response to reviewers' comments

We thank reviewers for their constructive evaluation of our manuscript. The main concern of all three reviewers was the inadequate citation of the work of Dobrev *et al* (2021). We agree that the credit given to the results of Dobrev *et al.* in our original manuscript was insufficient. We believe that our in-depth characterization of the Red1 complexes with Iss10, Ars2 and its dimerization, complements the Dobrev *et al.* study, and together these represent an important advance in our understanding of the structure and function of the RNA targeting complexes.

In the following point-by-point response to the reviewer's comments, we explain in detail how we implemented the necessary changes in the manuscript required to correct this omission. Importantly, we provide a series of new data and findings, in response to the points brought up by the reviewers.

Reviewer #1

In the fission yeast S. pombe, the MTREC complex plays a crucial role for targeting of transcripts including meiotic mRNAs and cryptic unstable transcripts. MTREC is considered to be a counterpart of the PAXT complex in human, and its core complex is composed of a zinc-finger protein Red1 and an RNA helicase Mtl1. In this manuscript, Foucher and colleagues investigate the interaction mode of Red1 with MTREC submodules. They solve the structure of the Red1-Iss10 complex, which is involved in the silencing of meiotic transcripts, by NMR. The crystal structure of the Red1-Ars1 complex is also determined. Red1 interacts with Ars2 through the "EDGE1" motif, and this interaction is conserved between ZFC3H1, a human counterpart of Red1 and hArs2.

Overall, the work is sound and well written. The experiments presented are convincing and well conducted. This study advances our understanding of the target recognition mechanism of RNA-targeting complexes not only in yeast but also in higher eukaryotes including human. I have a few minor comments listed below.

We thank the reviewer for these supportive comments on our manuscript.

Specific comments:

Point 1:

Lines 124-125: The authors state: "Red1, together with the Mtl1 helicase, has been proposed to form the MTREC core, but the molecular details of its interactions with the MTREC sub-modules remain unknown". However, a similar line of experiments has been reported by Dobrev et al. (doi: 10.1038/s41467-021-23565-3). The authors should compare the results with those of the previous study.

In order to better acknowledge the work of Dobrev *et al.* and to clarify that the focus of our study was on structural characterization of Red1 interactions with individual sub-modules, the first result section was shortened, and Figure 1c and Figure 1d were moved to Supplementary Figure 1. A reference to the work of Dobrev *et al.* was added. The paragraph was modified as follows (page 4, lines 125-134):

“Yeast two-hybrid analysis of Red1 interactions within MTREC

Red1, together with the Mtl1 helicase, has been shown to form the MTREC core^{6,7,14} (Fig. 1a), but the structural basis of its interactions with the MTREC sub-modules remains elusive. To understand how Red1 associates with its partners, we first performed Y2H assays to analyse its interactions with several MTREC members. We detected strong interactions with Ars2, Iss10 and Mtl1 as well as between Iss10 and Mmi1, that is in agreement with the recent Y2H analysis of the MTREC complex interactions¹⁴ (Fig. 1b, Supplementary Fig. 1a,b). In addition, our Y2H assay revealed an unexpected Red1 self-association (Fig. 1c). To obtain mechanistic insights into the MTREC/PAXT complex architecture, in this study, we performed structural and functional characterization of the Red1 interactions with Iss10 and Ars2 as well as of its self-association (Fig. 1d).“

Appropriate citations of the work of Dobrev *et al.* was introduced also in other sections as suggested by Reviewer 2 and 3 and are discussed below.

Point 2:

Figure 3a: Although Iss10 is unstable in red1-L205R mutant cells, it is detectable to some extent in total cell extracts. Is it possible to immunoprecipitate Iss10-GFP and to examine the interaction between Iss10 and Red1-L205R?

We have conducted the suggested experiment. The results further confirm that Red1 L205R point mutation compromises Red1 interaction with Iss10. The new data are mentioned in the Results section (page 6, line 193) and presented in panel of the Supplementary Figure 3a.

Point 3:

Figure 3c: Red1-L205R localization differs from wild-type Red1 localization in the iss10 deletion background, in which Red1 shows even distribution in the nucleus (Egan et al. doi:10.1261/rna.044479.114, Yamashita et al. doi:10.1093/nar/gkt763)

These results are indeed intriguing. Currently, we do not have a clear answer as to why they differ. We now add a sentence highlighting this point and speculate on what could be the most obvious reason. The following sentences are now added in the Discussion section (page 13, lines 450-457):

“We note that the mislocalization of Red1 in *red1-L205R* mutant cells, which causes Red1 to localize in an increased number of foci within the nucleus, differs from Red1 mislocalization in *iss10Δ* cells, in which Red1 shows even distribution in the nucleus^{6,26}. The reason for this difference is unclear but in *red1-L205R* cells the fact that Iss10 is still expressed may influence Red1 localization either due to residual direct interaction between Red1 and Iss10 or to the existence of an indirect interaction. In favour of the second possibility, affinity purifications of Red1 have suggested that Iss10-Mmi1 submodule may be connected to Red1 by several means⁶⁻⁸.”

Point 4:

It has been demonstrated that Ars2 regulates several genes by targeting upstream long non-coding RNAs. Is expression of such genes (e.g., pho1 and byr2) affected by the red1-E32R mutation?

We conducted the experiments to assess whether *red1-E32R* mutation impacts on the level of *pho1* and *byr2* mRNAs. The results of these experiments are that we do not see any significant changes in *pho1* and *byr2* mRNA levels in *red1-E32R* mutant cells. Noteworthy, these results are in agreement with a previous study showing that deletion of *red1* had no

significant impact on *pho1* and *byr2* mRNA levels (Thillainadesan *et al*, Nat Communications, 2020).

These additional experiments and results are now described in the Results section (page 9, lines 305-308), where the following was added: “It has been demonstrated that Ars2 regulates *pho1* and *byr2* protein-coding genes³⁷. We examined the possible effect of *red1-E32R* mutation on *pho1* and *byr2* mRNAs. However, no significant changes in *pho1* and *byr2* mRNA levels were observed between wild-type and *red1-E32R* mutant cells (Supplementary Fig. 8d).”

Point 5:

The significance of Red1 dimerization seems important, but remains ambiguous. How does non-dimeric L481R, I485R Red1 behave? Does it properly interact with Iss10 or Ars2? Does this mutation give an impact on the in vivo function of Red1?

We now have Y2H results with full-length proteins showing that the L481R, I485R Red1 double mutant essentially disrupts the Red1 dimerization (mutating one or both protomers) and that the double mutant does not impact binding of Iss10 and Ars2. These results were now added as panel f in Figure 7 and Supplementary fig. 13a. (page 12, lines 402-405)

We have also conducted the study of Red1 dimerization and its functional consequences on MTREC *in vivo*. Red1 dimerization was assessed by constructing strains co-expressing a Red1-HA₃ and a Red1-Myc₁₃. Co-immunoprecipitation (co-IP) experiments show that Red1 does indeed dimerize *in vivo*. The construction of strains co-expressing Red1-HA₃ together with Red1- Δ coiled-coil or Red1-L481R-I485R mutants and co-IPs show that both mutations compromise Red1 dimerization but do not completely disrupt the interaction. We made additional strains co-expressing Red1-HA₃ together with Red1-TAP and obtained similar results. In parallel, we investigated whether expression of these Red1 dimerization mutants resulted in loss of function. For that we examined cell growth and RNA levels of transcripts known to be degraded by MTREC. These investigations did not show any significant changes, indicating that the partial loss of dimerization imposed by these mutations does not lead to a loss of function under the growth conditions used for these experiments.

These new findings are now reported in the Results section (page 12, lines 406-418) as follow: “We then investigated whether Red1 dimerizes *in vivo*. We constructed strains co-expressing the proteins Red1-Myc₁₃ and Red1-HA₃. Co-IP experiments showed that indeed Red1 dimerizes within *S. pombe* (Fig. 7g). Co-expression of Red1-Myc₁₃ together with Red1- Δ CC-HA₃ (Δ coiled-coil corresponds to the deletion of residues 452-524) or Red1-L481R-I485R-HA₃ mutants showed that both mutations compromise Red1 dimerization but do not completely disrupt the interaction (Fig. 7g). We made other strains co-expressing Red1-HA₃ constructions together with Red1-TAP and obtained similar results (Supplementary Fig. 13b). We next explored whether the unique expression of either Red1 dimerization mutants resulted in Red1 loss of function. However, cell growth assays showed that *red1- Δ CC* or *red1-L481R-I485R* mutant cells growth is similar to wild-type cells (Supplementary Fig. 13c). In addition, examination of the RNA level of transcripts known to be degraded by MTREC showed no significant changes of their levels (Supplementary Fig. 13d). Thus, these results show that Red1 dimerizes *in vivo* and that the Red1 coiled-coil domain contributes to this dimerization in *S. pombe*.”

We also discuss these new results in the Discussion section (page 15, lines 518-529) as follows: “Finally, we also demonstrate the ability of Red1 to dimerize via its conserved coiled-coil region both *in vitro* and in *S. pombe*. Given that Iss10, Ars2 and Mtl1 can each bind to a single Red1, it is possible that the MTREC complex might exert its function as a dimeric assembly, where at least these subunits are present in pairs. This would be consistent

with the reported dimerization of Mmi1⁴⁹, which is a direct binder of meiotic transcripts in *S. pombe*^{20–22,50}. Mmi1 is dimerized via its interaction with its dimeric partner Erh1 and this is important for meiotic transcript degradation by MTREC⁴⁹. Functional analyses of Red1 mutants that lost the capacity to homodimerize *in vitro* and in Y2H assays did not reveal any significant loss of function for the *S. pombe* cells expressing these Red1 mutants. Importantly, our co-immunoprecipitation experiments showed that in *S. pombe* these Red1 mutants can still partially dimerize, indicating that *in vivo* Red1 can dimerize by at least one other mechanism. Better understanding of the role of Red1 dimerization within MTREC will thus require further characterization.”

Minor comments:

Point 6:

line 187: F215R should be Phe215.

This is now corrected in the text. (page 6, line 182)

Point 7:

Figure 2j: Gal4-AD (or BD?) should be Iss10.

“Gal4-AD” was corrected to “Gal4-BD: Iss10” in Figure 2j.

We realised, that there was a swap between BD and AD labelling in all Y2H figures. It is now corrected.

Point 8:

Figure 5f: Gal4-AD (or BD?) should be Ars2.

“Gal4-AD” was corrected to “Gal4-BD: Ars2” in Figure 5f.

Point 9:

line 337; A. thaliana SERRATE should be explained here.

To explain *A. thaliana* SERRATE, the sentence was modified as follows and a reference was added (page 9, lines 323-324) : “Similar differences are also observed when compared to the structure of the *A. thaliana* ARS2 ortholog known as SERRATE³⁸.” In the following text we then refer to SERRATE as ARS2/SERRATE.

Reviewer #2

In their manuscript “Structural analysis of Red1 as a conserved scaffold of the RNA-targeting MTREC/PAXT complex” Foucher and co-workers expand the structural and functional knowledge about the interaction network of the S. pombe MTREC complex and its human equivalent PAXT. While the core exosome and the subcomplexes TRAMP / NEXT have been comparably well studied, information about the human PAXT complex and the functionally homologous S. pombe MTREC complex is still incomplete with respect to their inner workings and functionality. This work focuses on the interactions of the central PAXT scaffolding protein Red1 with Iss10 and Ars2. Initially, the interaction network of Red1 within the MTRC complex is explored by Y2H assays partially complementing an earlier study. The authors provide first structural insight into the Red1-Iss10 interface by NMR. Also, they extend their prior structural study on human Ars2 now with the structure of S. pombe Ars2 in complex with Red1. The structural data now allows detailed comparison between the yeast and human complexes and rationalizes mutually exclusive interactions of hARS2 with various

“ED/EGEI/L” motif-possessing RNA regulators such as hZFC3H1, hFLASH or hNCBP3. Lastly, this study sheds light on a potential dimerization of the MTREC/PAXT complexes via the respective Red1 and hZFC3H1 proteins. It is noteworthy that this study nicely implements AlphaFold. In summary, structural and in vivo data on the RNA-targeting MTREC/PAXT complexes presented in this work provides further insight into the respective complex architecture and function.

The work is original, well and comprehensively written, and parts of it are of high interest. The experimental work is technically sound and provides strong evidence for the conclusions made. The authors provide further details on the interaction map of the S. pombe MTREC complex. However, a large fraction of these interactions has been reported before, several claims for novelty of the data are not justified. However, the structures as well as the wealth of in vivo data are novel and make a valuable contribution to the field. This work will seed and alleviate follow up structural studies of the MTREC complex and partially, this work may be transferred to the related human PAXT complex.

We thank the reviewer for appreciating the importance of our study.

In summary, I recommend a major revision of this article as a prerequisite for publication in Nature Communications. In particular, by the omission of published data by Dobrev et al., some results of this study are presented as novel which is not justified. An adequate citation of Dobrev et al. in the respective parts of this study and comparison has to be included to guarantee an adherence to good scientific practice.

We agree that work of Dobrev *et al.* was not appropriately cited in the original manuscript. Upon revision, we have now corrected this omission according to the reviewer comments.

Major points

Point 1:

Most of the Y2H interactions, e.g. Ars2-Red1 or Iss10-Red1 described on page 4, lines 125-135, have been already reported by Dobrev et al. (Nat. Commun 2021), which is cited as ref 14. This needs to be pointed out clearly and not sold as new data. Also, the role of Red1 as MTREC scaffolding protein has been proposed by Dobrev et al..

This point has been addressed in Point 1 of response to Reviewer 1.

In order to better acknowledge the work of Dobrev *et al.* the first result section was shortened, Figure 1c and Figure 1d were moved to Supplementary figure 1 and a reference to the work of Dobrev *et al.* was added.

Point 2:

Most of the Y2H interactions, e.g. between Red1 and Iss10 were already narrowed down by Dobrev et al. (Nat. Commun. 2021). Again, this needs to be pointed out clearly and not sold as new data. I see it very critical not to mention the study by Dobrev et al. here since this is out already for a while. The novelty advertised here is simply not present.

The “**Red1 interacts with the N-terminus of Iss10**” and “**NMR structure of the Red1-Iss10 complex**” sections were fused into a single “**NMR structure of the Red1-Iss10 complex**” section. The description of the interacting domain mapping was shortened and the results of Dobrev *et al.* cited as suggested.

This section was modified as follows (page 4, lines 137-146):

“In order to define the interacting regions between Red1 and Iss10 that would be suitable for structural analysis, we used Y2H that revealed a clear interaction between the N-terminal part of Iss10 (1-152) and the middle part of Red1 (138-390) (Supplementary Fig. 1c,d). The binding of these two fragments was confirmed *in vitro* by Strep-tag pull-down experiments,

requiring co-expression of the two proteins (Supplementary Fig. 1e, lane 4). Several deletions based on limited proteolysis and sequence alignments were produced, eventually yielding Red1¹⁹²⁻²³⁶ and Iss10¹⁻⁴⁵ constructs as minimal binding regions (Supplementary Fig. 1c, lane 6). Noticeably, similar domain boundaries of the two proteins required for the interaction were recently defined by Y2H assays¹⁴. Multi angle laser light scattering (MALLS) showed that these constructs form a homogenous complex sample with a 1:1 stoichiometry (Supplementary Fig. 1f). “

Point 3:

Page 6, line 199: Quoting “While red1Δ cells showed only a moderate growth defect on solid rich medium at 30°C...”. The prior characterization of this mutant by Dobrev et al. is omitted here. It has to be included and compared with this study.

We now have mentioned and added references to the previous studies showing that loss Red1 leads to a growth defect, which is exacerbated at low and high temperature (Sugiyama et al, EMBOJ, 2011 and Dobrev et al. 2021). (page 6, lines 195-198) and (page 9, lines 296-298)

Point 4:

Page 7, lines 217-231: Again, the authors sell their findings on the interface between Red1 and the short Ars2 motif as novelty. Although corroborated by gel filtration and ITC in this study, this interface was already published by Dobrev et al.

The “Red1 interacts with Ars2 via its conserved N-terminal motif “ and^[SEP] “Ars2-Red1 complex crystal structure determination” sections were now fused into a single “Crystal structure of the Ars2-Red1 complex” part. The description of the interacting domain mapping was shortened and the results of Dobrev *et al.* cited. The gel filtration profile of the minimal ARS2-Red1 complex was included into Figure 4a.

Page 7, lines 217-227

“To define the Red1 region interacting with Ars2, we used Y2H and gel filtration chromatography assays, showing a clear interaction between full-length Ars2 and Red1¹⁻¹⁴⁰ (Supplementary Fig. 4 a-c). Several further truncations of Red1¹⁻¹⁴⁰ were assayed, which eventually revealed that Red1²⁰⁻⁴⁰ is sufficient for Ars2 binding (Fig. 4a, Supplementary Fig. 4b-e). Interestingly, this Red1 fragment contains a short sequence motif “EDGEI”, which is significantly better conserved among several *Schizosaccharomyces* species than the rest of Red1¹⁻¹⁴⁰ (Supplementary Fig. 4f). To further characterize the interaction between Ars2 and this Red1 motif, which has also recently been shown to bind Ars2 by Dobrev *et al.*¹⁴, we used isothermal titration calorimetry (ITC). Both Red1¹⁻⁵⁴ and Red1²⁰⁻⁴⁰ (with and without MBP tag) bound Ars2 with an equivalent dissociation constant (K_d) of about 5 μM (Fig. 4b, Supplementary Fig. 5a,b), further supporting the hypothesis that Red1 region spanning residues 20-40 is a major determinant of the Ars2 binding.”

We also modified a following section mentioning previous mutagenesis of Red1 by Dobrev et al. (Page 8, lines 280-285):

“In addition, the role of E32R was supported by ITC measurements, since no binding to Ars2¹⁰⁷⁻⁵²⁷ was observed for the peptide containing the E32R mutation (Fig. 5e). In a previous study an E32A, D33A, E35A triple mutant was required to disrupt the binding between full length Red1 and Ars2¹⁴. Using Y2H assays, we could show that the Red1 E32R mutation is sufficient to prevent the interaction with Ars2 in the context of full-length proteins (Fig. 5f).”

Point 5:

Figure 4c was already shown by Dobrev et al. in a similar fashion. This is not indicated. Likewise, the observation that the N-terminal motif of Ars2 in yeast and human is similar was

already made in the earlier study.

We now fused the sections describing the conservation of the EDGEI motif and of the ARS2 Red1-binding surface into a single section following the Ars2-Red1 structure description called **“The Red1-Ars2 interface is conserved in human”**. The fact that the EDGEI motif is conserved in the sequence of ZFC3H1, reported by Dobrev *et al.*, is now cited. Panel 4c was removed. Panel 4d was moved to Figure 6 (now panel 6e). Panel 4e was removed. Previous panels 6e and 6f were now moved to Supplementary Figure 10 (now panels 10c and 10b, respectively). Finally, we added a new Panel 6f, highlighting the EDGEI motif present in several human RNA biogenesis factors. We also added a paragraph describing these EDGEI motif-containing proteins and previously described link of some of them to ARS2. These are now also discussed in the Discussion.

Page 10, lines 343-383

“The Red1-Ars2 interface is conserved in human

The Ars2 residues that interact with the Red1 EDGEI motif are generally very well conserved across species (Fig. 6c,d) suggesting that the interaction of Ars2 with this motif could be universal. The only exceptions are *S. pombe* L484 and L486, which are both replaced by lysine in most other species that likely retain hydrophobic contacts with I36 (Fig. 6c,d, Supplementary Fig. 10a). Correspondingly, it has already been reported that the EDGEI motif is conserved in the human PAXT subunit ZFC3H1, a putative human counterpart of Red1, implying that an analogous interaction between ARS2 and ZFC3H1 might exist within the PAXT complex¹⁴. To confirm this hypothesis, we used pull-down experiments with GST-tagged human ARS2¹⁴⁷⁻⁸⁷¹ and MBP-tagged ZFC3H1¹²⁻³³ which revealed a direct interaction between ARS2 and ZFC3H1 mediated by the conserved ZFC3H1¹²⁻³³ motif (Fig. 6e, lanes 4 and 7). Importantly, the E23R mutation analogous to the above-described E32R mutation in Red1 (Fig. 5d-f) severely reduced ZFC3H1¹²⁻³³ binding to hARS2 in an MBP-pull down assay (Supplementary Fig. 10b, lane 3). An I27L mutation did not perturb the binding (Supplementary Fig. 10b, lane 4), indicating that isoleucine in position 27 is not strictly required and can be replaced by a leucine. We could also show that a short peptide containing only 12 residues (ZFC3H1²⁰⁻³¹) is sufficient for the interaction with ARS2 (Supplementary Fig. 10b, lane 2). To further confirm the conservation of the Ars2-Red1 interface, we used MBP-pull down assays, where the GST-tagged human ARS2 (residues 147-871) bound equally well to MBP-Red1 or MBP-ZFC3H1 “EDGEI” motif-containing peptides (Supplementary Fig. 10c).

In addition to ZFC3H1, human ARS2 also interacts via a similar “EEGEI” motif with FLASH, a large protein involved in histone mRNA biogenesis^{32,40}. FLASH⁹³¹⁻⁹⁴³ binds ARS2¹⁴⁷⁻⁸⁷¹ with a K_d of 0.5-5 μ M depending on salt concentration³², reminiscent of the Red1-Ars2 binding. Accordingly, in the human system, a K719A, K722, K734A triple mutant located at the EDGEI motif-binding surface of ARS2 (Supplementary Fig. 10a) abolished the hARS2-FLASH interaction³².

A search for other human proteins containing possible ARS2-binding motifs with SlimSearch⁴¹, identified several RNA biogenesis factors (Fig. 6f). These include three other Zn-finger proteins - ZC3H4, ZC3H6 and ZC3H18. Similarly to ZFC3H1, they contain two or three tandem copies of the motif. The strong effect of the single E23R mutation in one of the ZFC3H1 motifs (Supplementary Fig. 10b, lane 4) might indicate that individual motifs interact with different affinities depending on surrounding residues. NCBP3, linked to mRNA export⁴² also possesses two separated copies of the motif. Notably, NCBP3 was shown to directly bind ARS2 and the C-terminal leg of ARS2 was required for this interaction³². Finally, THRAP3, PRPF4B, RTF1, and CPSF2 contain one motif. In all these proteins, the conserved ARS2-binding motif is located in disordered regions as judged using AlphaFold2 predicted models. Importantly, ZFC3H1, ZC3H4, NCBP3, THRAP3, PRPF4B and CPSF2 were all found enriched in affinity purification/mass spectrometry (AP-MS) analysis with

native hARS2 compared to the hARS2 K719A, K722A, K734A triple mutant in the ARS2 C-terminal leg³².

Together, these results show that the *S. pombe* Ars2-Red1 interface is well conserved among species. From these results we also hypothesise, that in addition to hZFC3H1 and FLASH, the ARS2 ED/EGEI/L binding surface is involved in connecting CBC-ARS2 bound-RNAs to several other regulators of RNA metabolism, likely in a mutually exclusive way.“

Point 6:

Page 8, lines 275 – 281: I suggest to complement these findings with a PISA server analysis of the interface (or parts of it) to maybe obtain more information about physiological relevance.

We now including a sentence with the PISA server analysis (page 8, lines 252-254):

“Analysis of the extensive dimer interface with PDBePISA server³⁶ at the EBI revealed that it buries 4946 Å² on each Ars2 protomer, including 37 hydrogen bond and 18 salt bridge contacts between the two molecules. “

Point 7:

I would recommend to assess the effects of Red1 dimerization also in vivo based on the L481R, I485R mutants where dimer formation is abolished in vitro. It would be interesting to see a functional relevance or rule that out.

We have now conducted a series of experiments addressing these points. They are described in Point 5 of response to Reviewer 1, whom had similar comments.

Minor points

Point 8:

The abstract is somehow missing a rational why this study was conducted.

The word count of the abstract is limited to 150 words. We think that the rational for conducting this study is explained in the last sentence:

“Our results, combining structures of three Red1 interfaces with *in vivo* studies, provide mechanistic insights into conserved features of MTREC/PAXT architecture.”

Point 9:

Page 4, line 114 and Page 4, line 118: Rather use present, “demonstrate” instead of “demonstrated”

This is now corrected in the text. (page 4, line116)

Point 10:

In Figure 2, f - h, the side chains of Iss10 should be coloured differently to avoid confusion with the Red1 backbone.

The side chains in Figure 2f-h were now coloured in a more distinctive colour.

Point 11:

In Figure 2 j, the fusion partner of the GAL-4-BD construct is missing.

This is now corrected in the figure. The same was corrected also in Figure 5f.

Point 12:

Page 6, line 179: “To this end, we mutated Ala198...” does not provide information to which residue it was mutated. The same for both hydrophobic residues. What is the rationale behind these mutations?

To provide this missing information, the text was modified as follows (page 5, lines 175-177):
“To this end, we generated an A198E mutation in the centre of the coiled-coil formed with Iss10 and two L205R and F215R mutations that we hypothesised should directly disrupt the interaction with Iss10 helix α 1 (Fig. 2f-h).”

To clarify the reason why these mutations were produced we modified the preceding sentence as follows (page 5, lines 173-175):

“In order to assess of the role of the Red1-Iss10 interaction *in vivo*, we were then interested in identifying Red1 mutations that could disrupt its interaction with Iss10 without affecting the Red1 HTH fold.”

Point 13:

Page 6, line 184: The authors may consider using one-letter-code for amino acids instead of three-letter-codes, however, not both. Figures come along less crowded with one-letter-coded amino acids.

As suggested, we modified all the amino acid labelling to on-letter-code in the text and figures.

Point 14:

Page 6, line 186: replace “purify on” by “purify employing”

This is now corrected in the text. (page 6, line 182)

Point 15:

Page 6, line 187: “using an Y2H assay”

This is now corrected in the text. (page 6, line 183)

Point 16:

Page 6, lines 188-190: Quoting “Ultimately, the L205R mutation was selected for in vivo studies (see below) and the integrity of mutated Red1192-236 was verified by gel filtration, where its elution profile was similar to the WT protein (data not shown)”. I would not really call it “verification” without showing the WT gel filtration data.

Upon revision, the gel filtration profiles were added to Supplementary Figure 2g. (page 6, lines 185-186)

Point 17:

Page 6, line 194: Replace “In order to evaluate in vivo the function of Red1-Iss10 binding interface” by “In order to evaluate in vivo the function of the Red1-Iss10 binding interface”

This is now corrected in the text. (page 6, line 190)

Point 18:

Page 7, line 214: rather “the Red1-Iss10 binding interface”

This is now corrected in the text. (page 6, line 210)

Point 19:

In figure 3, c,d, panels have different sizes. Figure size has to be increased - at the present panel size, contents are not comprehensively visualized.

The changes have been made accordingly.

Point 20:

In figure 3 f, error bars have to be coloured differently and should be in front of the bars for visibility.

We changed the colour of the error bars to make them more visible.

Point 21:

Page 9, lines 297 – 298: Quoting: “The integrity of the Ars2 mutants was verified by gel filtration, with an elution profile similar to the WT protein (data not shown).” I recommend to include WT gel filtration data.

Upon revision, the gel filtration profiles were added to Supplementary Figure 7c. (page 8, lines 274-276)

Point 22:

In Figure 5d, lane 6, the wt seems to bind sub-stoichiometrically? Is there an explanation?

We don't have an explanation for this, other than possible fast interaction kinetics and loss of Red1 during the washing step of the pull-down which is necessary to eliminate non-specific binding of Red1 to the resin. In our experimental conditions, pull-down analysis of the *S. pombe* Red1-Ars2 interaction always works like this. As shown, in the novel Figure 6e, it works better with human proteins. We now repeated the pull down of Figure 5d several times, using large excess of Red1 and purifying Ars2 first on gel filtration to remove possible aggregates, but still the binding is sub-stoichiometric. We now used a new gel image in Figure 5d, which seems a bit better. The lack of binding of the E32 Red1 mutant was also confirmed by ITC, Y2H and co-IP in *S. pombe* cells.

Point 23:

Page 9, line 313: Rather “impact of the red1-E32R point mutation”

This is now corrected in the text. (page 9, line 292)

Point 24:

Page 9, line 323: Rather “As MTREC degrades a specific population”

This is now corrected in the text. (page 9, line 301)

Point 25:

Page 10, lines 326-329: “More diffused dots” cannot be reasoned from Figure 5i in my opinion.

We have reformulated the text as follows (page 9, lines 309-311): “In cells expressing the Red1-E32R-TAP mutant protein, Ars2-GFP signal also formed nuclear foci like in wild-type cells but they tended to be more difficult to distinguish from the more even nuclear GFP signal when compared to wild-type cells (Fig. 5i).”

We also conducted a statistical analysis with more than 100 nuclei per replicate, and with two different biological replicates. Quantification of the intensity of the signal within each nucleus was performed. Interestingly, the new data show a significant reduction of Ars2-GFP nuclear signal in *red1-E32R* mutant cells compared to wild-type cells. Since the level of Ars2-GFP is the same in wild-type and *red1-E32R* mutant cells (Fig. 5g), these results suggest that part of Ars2-GFP population diffuse to the cytoplasm in the mutant cells. We have added the following sentences describing this finding in the Results section (page 9, lines 312-317): “Interestingly, quantification of the intensity of the signal within each nucleus showed a reduction of Ars2-GFP nuclear signal in *red1-E32R* mutant cells compared to wild-type cells, suggesting that some of Ars2-GFP proteins diffuse to the cytoplasm (Supplementary Fig. 8e). The localization of Red1-TAP in nuclear foci did not significantly change (Supplementary Fig. 8f,g). Thus, *in vivo*, Red1-Ars2 binding interface is critical for Red1-Ars2 interaction, for optimal cell growth and nuclear localization of Ars2 where Red1 and other subunits of MTREC localize.” Additionally, we have replaced the images of Figure 5i to better illustrate this new data, and reported the nuclei counting (box plot, Supplementary Figure 8e).

Point 26:

Page 10, line 337: *A. thaliana* SERRATE has to be introduced to the readers first before comparing it with Ars2.

The same suggestion as by Reviewer 1. To explain *A. thaliana* SERRATE, the sentence where SERRATE is mentioned first was modified as follows and a reference added (page 9, line 323-324): “Similar differences are also observed when compared to the structure of the *A. thaliana* ARS2 ortholog known as SERRATE³⁸.” In the following text we then refer to SERRATE as ARS2/SERRATE. (page 10, line 338)

Point 27:

Page 13. Lines 462-463: This has also been implied by Dobrev et al.

Upon revision, this section has been rephrased (page 14, lines 489-492):

“We show that the C-terminal leg interacting surface is conserved between *S. pombe* and human ARS2. Similarly, the Red1 EDGEI motif is conserved in human hZFC3H1¹⁴, a subunit of the PAXT complex. Correspondingly, we show that hARS2 and hZFC3H1 interact and the E23R mutation in hZFC3H1 (equivalent to E32R in Red1) is sufficient to disrupt the complex.”

Point 28:

The discussion may benefit from an overview figure that goes maybe even beyond fig 7f wr to complexity, e.g. extending it towards a hypothetical PAXT assembly.

We now included an extra panel, showing also the hypothetical PAXT complex model (Figure 8b). The *S. pombe* MTREC model was moved to Figure 8a.

Dobrev *et al.* did not validate the hypothesis, based on conservation of interacting residues, that in human MTR4 interact with ZFC3H1 with equivalent regions as in *C. thermophilum*. We now used AlphaFold2 to predict the human complex structure revealing the interaction of the expected the MTR4-binding region of ZFC3H1 which in addition also packs against a small downstream helical domain. Since, it is likely that the complete MTR4-binding domain spans both regions, we included these models in a new Supplementary Figure 15. An equivalent domain was predicted also for *S. pombe* Red1 (AF-Q9UTR8, residues 575-703).

Reviewer #3

In their manuscript the authors mapped interactions between components of the MTREC complex and solved structure of the two peptides derived from Red1 and Iss10 as well as structure of fragments of Red1 and Ars2.

The data is of decent quality and the findings are interesting. However, I am appalled by the fact that most of the presented data has been published in Nat. Communications 9 months ago. The authors mention this manuscript, but (intentionally) fail to tell the reader that their findings are not novel. It can very well be that the authors got badly scooped, but to plainly pretend that the data in the literature is not there is not acceptable, especially since the time between the ref 14 paper and today is significant (~9 months). As such, I cannot recommend the current manuscript for publication. I suggest that the authors remove the known parts and write a manuscript for submission in a more specialized journal that focus on the novel structural data and the functional assays only.

We thank the reviewer for his comments, appreciating the quality and interest of our findings. We agree that the work of Dobrev *et al.* was not properly acknowledged and cited. Upon revision this has now been corrected, by following advice of all three reviewers.

The clear neglecting of the data in Ref 14 is clear from e.g. the following statements:

Point 1:

The subunit composition of the S. pombe MTREC and human PAXT complexes has been characterized by proteomics”. This sentence discredits previous work by the Sinning and Fischer labs (ref 14) that have made use of Y2H and Y3H approaches and those labs revealed a detailed analysis of MTREC submodule interactions.

In our opinion this sentence does not discredit the previous work of Dobrev *et al.* as it is related to proteomics studies that defined the subunit composition of both MTREC and PAXT complexes (References 6-9,11) which was not done in the study of Dobrev *et al.* We have now included a direct reference to the 2YH work of Dobrev *et al.* in the following sentence (see below).

Point 2:

“In MTREC, the Red1 protein was suggested to form a scaffold of the complex, interacting with multiple subunits and linking individual sub-modules together^{6,7,14}”. These previous data do not only suggest something, the unambiguously show an important role of Red1.

We now modified the sentence as follows (page 3, lines 72-74):

“Recently, using yeast two-hybrid (2YH) and pull-down assays, the Red1 protein was shown to form a scaffold of the MTREC complex, interacting with multiple subunits and linking individual sub-modules together¹⁴.

Point 3:

“. Red1 interacts with the Mtl1 Arch domain via its zinc-finger containing domain at its C-terminus¹⁴”. This has been mapped in ref 14 accurately to Red1 (Ars2, Iss10, Pla1, Rmn1).

This sentence was referring to the crystal structure of the Red1-Mtl1 complex published by Dobrev *et al.* It does not concern any other MTREC subunits.

The sentence was anyway modified to be clearer (page 3, lines 75-77):

“A crystal structure of a minimal Red1-Mtl1 complex, revealed how Red1 interacts with the Mtl1 Arch domain via its zinc-finger containing domain at its C-terminus¹⁴.”

Point 4:

“How the Red1-Mtl1 core interacts with the individual MTREC sub-modules and what is their function in MTREC/PAXT activity, however, remains unclear”, data is in ref 14 paper.

The sentence was modified as follows (Page 3, lines 80-82):

“However, there is no 3D structural information available on the interactions that the Red1-Mtl1 core forms with the individual MTREC sub-modules. Similarly, the function of these modules in MTREC/PAXT activity still remains unclear.”

The whole introduction paragraph now reads as this (Page 3, lines 71-82):

“The subunit composition of the *S. pombe* MTREC and human PAXT complexes has been characterized by proteomics^{6-9,11}. Recently, using yeast two-hybrid (2YH) and pull-down assays, the Red1 protein was shown to form a scaffold of the MTREC complex, interacting with multiple subunits and linking individual sub-modules together¹⁴. Red1 consists of 711 residues (compared to the 1989 residues of its ZFC3H1 counterpart in human PAXT) and is predicted to be mostly disordered. A crystal structure of a minimal Red1-Mtl1 complex, revealed how Red1 interacts with the Mtl1 Arch domain via its zinc-finger containing domain at its C-terminus¹⁴. In human and *S. cerevisiae*, Mtl1 counterpart MTR4 has been structurally characterised¹⁵ revealing molecular details of its interactions with its partners including the NEXT subunit ZCCHC8¹⁶ and the RNA exosome to whom MTR4 presents the targeted RNA for degradation^{17,18}. However, there is no 3D structural information available on the interactions that the Red1-Mtl1 core forms with the individual MTREC sub-modules. Similarly, the function of these modules in MTREC/PAXT activity still remains unclear”

Point 5:

“How the CBC-ARS2 complex is integrated is currently unknown. “. A clear motif has been identified in Ref. 14.

The sentence was modified as follows (page 4, lines 109-111):

“The atomic details underlying how the CBC-ARS2 complex is recruited to the exosome-linked RNA targeting complexes and what role this module plays are poorly understood.“

Point 6:

“. We mapped minimal Red1 regions required for its interaction with Iss10 and Ars2, respectively.” This was done before in Ref 14.

The sentence was removed. (page 4, line 113)

Point 7:

In my opinion also many paragraphs in the results section do not provide new data or insights, inducing: “Red1 forms multiple interactions within the MTREC complex “, “Red1 interacts with the N-terminus of Iss10 “, “Red1 interacts with Ars2 via its conserved N-terminal motif” and many parts of the other sections.

We removed some of these subheadings, shortened the paragraphs concerned with interacting domain mapping and acknowledged work of Dobrev *et al*, when appropriate. This is described in more detail in response to Reviewer 2 – Points 1, 2, 4 and 5.

Point 8:

The structure of the complex of the Red1 and Iss10 peptide is novel, but the interacting residues had been determined before. The structure of the Ars2 and a Red1 N-terminal

peptide appears novel. The PDB validation report, however, clearly state “This wwPDB validation report is NOT for manuscript review”.

The interacting residues of Red1 and Iss10 complex were not determined before. Dobrev *et al*, mapped interacting regions by Y2H. They did not determine interacting residues nor structure.

The PBP validation report was provided according to the submission guidelines of Nature Communications.

Reviewers' Comments:

Reviewer #1:

Remarks to the Author:

The authors have satisfactorily addressed concerns by this reviewer and have made the appropriate amendments. I support publication of the manuscript.

Typos that should be corrected before publication:

line 143: Supplementary Fig. 1c should be 1e.

line 923: in conserved should be is conserved.

Reviewer #2:

Remarks to the Author:

The authors have now markedly improved their manuscript and have settled most of my concerns. However, a few major points (in bold and italics) still remain. In summary, I find it disappointing that despite the justified criticism of all three referees the authors still appear to have a hard time to acknowledge the prior study by Dobrev et al. appropriately (see below). Again, the deliberate omission of prior results to "enhance" a manuscript is bad scientific practice.

Point 4:

Page 7, lines 217-231: Again, the authors sell their findings on the interface between Red1 and the short Ars2 motif as novelty. Although corroborated by gel filtration and ITC in this study, this interface was already published by Dobrev et al.

The "Red1 interacts with Ars2 via its conserved N-terminal motif" and ^{Fig. 1} "Ars2-Red1 complex crystal structure determination" sections were now fused into a single "Crystal structure of the Ars2-Red1 complex" part. The description of the interacting domain mapping was shortened and the results of Dobrev et al. cited. The gel filtration profile of the minimal ARS2-Red1 complex was included into Figure 4a.

Page 7, lines 217-227

"To define the Red1 region interacting with Ars2, we used Y2H and gel filtration chromatography assays, showing a clear interaction between full-length Ars2 and Red11-140 (Supplementary Fig. 4 a-c). Several further truncations of Red11-140 were assayed, which eventually revealed that Red120-40 is sufficient for Ars2 binding (Fig. 4a, Supplementary Fig. 4b-e). Interestingly, this Red1 fragment contains a short sequence motif "EDGEI", which is significantly better conserved among several *Schizosaccharomyces* species than the rest of Red11-140 (Supplementary Fig. 4f). To further characterize the interaction between Ars2 and this Red1 motif, which has also recently been shown to bind Ars2 by Dobrev et al.¹⁴, we used isothermal titration calorimetry (ITC). Both Red11-54 and Red120-40 (with and without MBP tag) bound Ars2 with an equivalent dissociation constant (K_d) of about 5 μM (Fig. 4b, Supplementary Fig. 5a,b), further supporting the hypothesis that Red1 region spanning residues 20-40 is a major determinant of the Ars2 binding."

We also modified a following section mentioning previous mutagenesis of Red1 by Dobrev et al. (Page 8, lines 280-285):

"In addition, the role of E32R was supported by ITC measurements, since no binding to Ars2107-527 was observed for the peptide containing the E32R mutation (Fig. 5e). In a previous study an E32A, D33A, E35A triple mutant was required to disrupt the binding between full length Red1 and Ars214. Using Y2H assays, we could show that the Red1 E32R mutation is sufficient to prevent the interaction

with Ars2 in the context of full-length proteins (Fig. 5f)."

Again, Dobrev et al. have already shown the existence of the EDGEI motif, also for Red1. Quoting the authors, "Interestingly, this Red1 fragment contains a short sequence motif "EDGEI", which is significantly better conserved among several Schizosaccharomyces species than the rest of Red11-140 (Supplementary Fig. 4f)." still implies that this motif is their finding, irrespective of an earlier corresponding mention of the Dobrev study in their manuscript. The proper context should be restored also here by mentioning the findings of Dobrev et al. prior to their own. In other words – the earlier findings by Dobrev are confirmed by the present study, and not vice versa. Likewise, in other instances throughout the improved manuscript, the authors make the earlier study by Dobrev et al. deliberately fall in the "also ran" category, by mentioning it after their own results (which still makes those appear novel to some degree). Other examples are, e.g., in point 2 where "Noticeably, similar domain boundaries of the two proteins required for the interaction were recently defined by Y2H assays14". comes after the description of own results. The same for point 27. I understand that these changes will 'water down' the findings of the current study, however, in my opinion the current approach is still not good scientific practice.

Point 6:

Page 8, lines 275 – 281: I suggest to complement these findings with a PISA server analysis of the interface (or parts of it) to maybe obtain more information about physiological relevance.

We now including a sentence with the PISA server analysis (page 8, lines 252-254):

"Analysis of the extensive dimer interface with PDBePISA server36 at the EBI revealed that it buries 4946 Å² on each Ars2 protomer, including 37 hydrogen bond and 18 salt bridge contacts between the two molecules. "

The PISA server also scores the interfaces for their physiological relevance. Maybe this should be included, too. These values given don't tell much, except that the interface is extended.

Point 8:

The abstract is somehow missing a rational why this study was conducted.

The word count of the abstract is limited to 150 words. We think that the rational for conducting this study is explained in the last sentence:

"Our results, combining structures of three Red1 interfaces with in vivo studies, provide mechanistic insights into conserved features of MTREC/PAXT architecture."

What I meant is that there is no scientific question why this study was conducted. E.g. "Inner workings of the PAXT complex still miss sufficient detail." or alike.

Point 28:

The discussion may benefit from an overview figure that goes maybe even beyond fig 7f wr to complexity, e.g. extending it towards a hypothetical PAXT assembly.

We now included an extra panel, showing also the hypothetical PAXT complex model (Figure 8b). The *S. pombe* MTREC model was moved to Figure 8a.

Dobrev et al. did not validate the hypothesis, based on conservation of interacting residues, that in human MTR4 interact with ZFC3H1 with equivalent regions as in *C. thermophilum*. We now used AlphaFold2 to predict the human complex structure revealing the interaction of the expected the MTR4-binding region of ZFC3H1 which in addition also packs against a small downstream helical

domain. Since, it is likely that the complete MTR4-binding domain spans both regions, we included these models in a new Supplementary Figure 15. An equivalent domain was predicted also for *S. pombe* Red1 (AF-Q9UTR8, residues 575-703).

Supplementary figure 15 is not referenced in the main text. And I would not use the term 'revealing' for an AlphaFold2 model, in particular, when it comes to side chain interactions.. It is merely an indication

Reviewer #3:

Remarks to the Author:

The revised version of the manuscript did not change initial concerns that were raised by me and one of the other reviewers: previously published data has not been acknowledged appropriately. In addition, I feel that the paper is in part not focussed enough. Three different topics or interactions (Red1-Iss10, Red1-Ars2 and Red1-Red1) that are presented are clearly related, but there are no real connections between the three parts. As a result, the paper fails a clear message. Finally, the solved structures are very standard and do not provide exciting or state-of-the-art insights. I suggest publishing the work in a more focused journal, e.g. NAR or RNA. The impact of the manuscript appears too limited to warrant publication in Nat. Com.

My conclusion is based on the following remarks:

The NMR structure is new. Considering the very small size of this complex it can not be considered technically challenging though. I wonder what an alpha-fold prediction would give. The authors did quite some alpha-fold and are thus able to address this.

Line 143-144: 'Noticeably, similar domain boundaries of the two proteins required for the interaction were recently defined by Y2H assays'. Why is this noticeable? It is just a repeat of previous experiments. In case the authors learned anything new they should highlight that. In case nothing new is learned they should just clearly state that they repeated the experiments that have been published before. (See top green arrow Fig 2C of ref 14).

Line 157-158: "While Red1 DNA binding has not been reported and no unspecific DNA binding was observed during Red1 purification, DNA interaction cannot be excluded.". It is up to the authors to exclude (or at least test) this, e.g. using NMR titration experiments.

Line 174-175: "identifying Red1 mutations that could disrupt its interaction with Iss10 without affecting the Red1 HTH fold.". The authors should show that their mutations (L205R and F215R) indeed do not destabilize the HTH fold. They should e.g. provide NMR spectra of the mutant proteins. In case the HTH fold is destabilized by the mutations, all in vivo effects could be due to an unfolded HTH domain and not be due to the disrupted interaction between the 2 proteins.

Line 223-224: "To further characterize the interaction between Ars2 and this Red1 motif, which has also recently been shown to bind Ars2 by Dobrev et al.14, ". The authors still fail to acknowledge previous data appropriately. In ref 14 the interaction between Ars2 and Red1 has been studied in quite some detail. The findings presented here just recapitulate published findings. The ignorance of the authors towards published data from others is further obvious from the statement "further supporting the hypothesis that Red1 region spanning residues 20-40 is a major determinant of the Ars2 binding.". Ref 14 clearly identified this region and even presents point mutations that disrupt the interaction Red1-Ars2 interaction.

Line 244-246: " The structure was determined by molecular replacement using the AlphaFold2 Ars2 model (AF-O94326) and refined to an Rfree of 30.2% and an Rworkof 24.5% (Supplementary Table

2).". The fact that the alpha-fold model can be used for molecular replacement raises the question how different the alpha-fold model is from the determined structure. The authors should report an overlay of the model and the structure. In case both are highly similar then I would say that it is fair to ask what the added value of the structure is. Clearly, the structure is a complex, but maybe alpha-fold is also able to predict the structure of the complex well.

The "Dimerization of Red1 C-terminus" does not provide any (functional) insights. A mutant that prevents the dimerization is identified, but the role of dimerization (if any exists) remains unclear.

Response to reviewers' comments

Reviewer #1

The authors have satisfactorily addressed concerns by this reviewer and have made the appropriate amendments. I support publication of the manuscript.

We are happy to read that the revised manuscript satisfactorily addressed reviewer's concerns. We thank the reviewer for constructive suggestions, which we believe improved the manuscript.

Typos that should be corrected before publication:

Point 1:

line 143: Supplementary Fig. 1c should be 1e.

The paragraph has been modified (see below) and this typo corrected (page 4, lines 135-140).

Point 2:

line 923: in conserved should be is conserved.

This error is now corrected (page 27, line 919).

Reviewer #2

The authors have now markedly improved their manuscript and have settled most of my concerns. However, a few major points (in bold and italics) still remain. In summary, I find it disappointing that despite the justified criticism of all three referees the authors still appear to have a hard time to acknowledge the prior study by Dobrev *et al.* appropriately (see below). Again, the deliberate omission of prior results to “enhance” a manuscript is bad scientific practice.

We thank the reviewer for all his/her previous and new comments. We agree that we did not acknowledge the work of Dobrev *et al.* properly. We sincerely apologize for this and we now corrected the text as suggested.

Point 1:

Point 4:

Page 7, lines 217-231: Again, the authors sell their findings on the interface between Red1 and the short Ars2 motif as novelty. Although corroborated by gel filtration and ITC in this study, this interface was already published by Dobrev *et al.*

The “Red1 interacts with Ars2 via its conserved N-terminal motif “ and “Ars2-Red1 complex crystal structure determination” sections were now fused into a single “Crystal structure of the Ars2-Red1 complex” part. The description of the interacting domain mapping was shortened and the results of Dobrev *et al.* cited. The gel filtration profile of the minimal ARS2-Red1 complex was included into Figure 4a.

Page 7, lines 217-227

“To define the Red1 region interacting with Ars2, we used Y2H and gel filtration chromatography assays, showing a clear interaction between full-length Ars2 and Red11-140 (Supplementary Fig. 4 a-c). Several further truncations of Red11-140 were assayed, which eventually revealed that Red120-40 is sufficient for Ars2 binding (Fig. 4a, Supplementary Fig. 4b-e). Interestingly, this Red1 fragment contains a short sequence motif “EDGEI”, which is significantly better conserved among several *Schizosaccharomyces* species than the rest of Red11-140 (Supplementary Fig. 4f). To further characterize the interaction between Ars2 and this Red1 motif, which has also recently been shown to bind Ars2 by Dobrev *et al.*¹⁴, we used isothermal titration calorimetry (ITC). Both

Red1-54 and Red120-40 (with and without MBP tag) bound Ars2 with an equivalent dissociation constant (Kd) of about 5 μ M (Fig. 4b, Supplementary Fig. 5a,b), further supporting the hypothesis that Red1 region spanning residues 20-40 is a major determinant of the Ars2 binding.”

We also modified a following section mentioning previous mutagenesis of Red1 by Dobrev et al. (Page 8, lines 280-285):

“In addition, the role of E32R was supported by ITC measurements, since no binding to Ars2107-527 was observed for the peptide containing the E32R mutation (Fig. 5e). In a previous study an E32A, D33A, E35A triple mutant was required to disrupt the binding between full length Red1 and Ars214. Using Y2H assays, we could show that the Red1 E32R mutation is sufficient to prevent the interaction with Ars2 in the context of full-length proteins (Fig. 5f).”

Again, Dobrev et al. have already shown the existence of the EDGE1 motif, also for Red1. Quoting the authors, “Interestingly, this Red1 fragment contains a short sequence motif “EDGE1”, which is significantly better conserved among several Schizosaccharomyces species than the rest of Red1-140 (Supplementary Fig. 4f).” still implies that this motif is their finding, irrespective of an earlier corresponding mention of the Dobrev study in their manuscript. The proper context should be restored also here by mentioning the findings of Dobrev et al. prior to their own. In other words – the earlier findings by Dobrev are confirmed by the present study, and not vice versa. Likewise, in other instances throughout the improved manuscript, the authors make the earlier study by Dobrev et al. deliberately fall in the “also ran” category, by mentioning it after their own results (which still makes those appear novel to some degree). Other examples are, e.g., in point 2 where “Noticeably, similar domain boundaries of the two proteins required for the interaction were recently defined by Y2H assays¹⁴”. comes after the description of own results. The same for point 27. I understand that these changes will ‘water down’ the findings of the current study, however, in my opinion the current approach is still not good scientific practice.

We agree with the reviewer and we now introduced additional modifications into the text:

Introduction:

The introduction paragraph now contains two references to the work of Dobrev *et al.*:

1: (page 3, lines 68-69)

“The Red1 protein was shown to form a scaffold of the MTREC complex, interacting with multiple subunits and linking individual sub-modules together¹⁴.”

2: (page 3, lines 75-80)

“Recently, using yeast two-hybrid (2YH) and pull-down assays Dobrev *et al.* defined the minimal interacting regions involved in the interactions of Red1 with Iss10 of the Iss10-Mmi1 sub-module and with Ars2 of the cap-binding complex¹⁴. However, there is no 3D structural information available on the interactions that the Red1-Mtl1 core forms with the individual MTREC sub-modules. Similarly, the function of these modules in MTREC/PAXT activity still remains unclear.”

Yeast two-hybrid analysis of Red1 interactions within MTREC (previously Point 1)

This part was modified accordingly (page 4, lines 127-130):

“In agreement with the recent Y2H analysis of the MTREC complex interactions¹⁴, we detected strong interactions with Ars2, Iss10 and Mtl1 as well as between Iss10 and Mmi1 (Fig. 1b, Supplementary Fig. 1a,b). In addition, our Y2H assay revealed an unexpected Red1 self-association (Fig. 1c).”

NMR structure of the Red1-Iss10 complex (previously Point 2):

This section was further shortened and the order of our and Dobrev *et al.* results inverted (page 4, lines 135-140):

“A Y2H and GST pull-down analysis recently revealed that Red1¹⁸⁷⁻²³⁶ and Iss10¹⁻⁵¹ represent the interacting regions of the complex Red1 forms with Iss10¹⁴. Our characterization of the complex using Y2H, Strep-tag pull-down and limited proteolysis experiments identified equivalent fragments Red1¹⁹²⁻²³⁶ and Iss10¹⁻⁴⁵ (Supplementary Fig. 1c-e). Multi angle laser light scattering (MALLS) showed that these constructs form a homogenous complex sample with a 1:1 stoichiometry (Supplementary Fig. 1f).”

Crystal structure of the Ars2-Red1 complex (previously Point 4):

This section was also further shortened and modified according to the reviewer's suggestion (page 7, lines 212-219):

“The interaction between Ars2 and Red1 has recently been shown, by Y2H and pull-down assays, to be mediated by a short conserved “EDGEI” motif of Red1¹⁴. Using Y2H and gel filtration chromatography we confirmed the implication of the EDGEI motif in the interaction with full-length Ars2, identifying a slightly longer construct Red1²⁰⁻⁴⁰ (Fig. 4a, Supplementary Fig. 4). To further characterize the interaction between Ars2 and this Red1 motif, we used isothermal titration calorimetry (ITC). Both Red1¹⁻⁵⁴ and Red1²⁰⁻⁴⁰ (with and without MBP tag) bound Ars2 with an equivalent dissociation constant (K_d) of about 5 μ M (Fig. 4b, Supplementary Fig. 5a,b), confirming that additional sequences surrounding EDGEI motif do not enhance the binding affinity for Ars2.”

Dobrev *et al.* also showed by Y2H that the C-terminal leg (450-516) of Ars2 can bind Red1. This result was not confirmed by pull-downs, where only the FL Ars2 was used. We also tried producing equivalent constructs in hARS2 but they did not seem stable. We believe that the C-terminal leg is meant to be embedded within the Ars2 structure to be fully functional. We now added this sentence (page 7, lines 221-223):

“Ars2 residues 450-516 were shown to bind Red1 by Y2H¹⁴, but in our hands short constructs encompassing this region of hARS2 did not produce stable protein.”

We also accordingly modified the following in “**The Red1-Ars2 interface is conserved in human**” section (page 10, lines 340-349):

“It has been reported that the EDGEI motif is conserved in the human PAXT subunit ZFC3H1, a putative human counterpart of Red1, implying that an analogous interaction between ARS2 and ZFC3H1 might exist within the PAXT complex¹⁴. Correspondingly, the Ars2 residues that interact with the Red1 EDGEI motif are generally very well conserved across species (Fig. 6c,d) suggesting that the interaction of Ars2 with this motif could be universal. The only exceptions are *S. pombe* L484 and L486, which are both replaced by lysine in most other species that likely retain hydrophobic contacts with I36 (Fig. 6c,d, Supplementary Fig. 10a). To assess the conservation of the interface in human PAXT, we used pull-down experiments with GST-tagged human ARS2¹⁴⁷⁻⁸⁷¹ and MBP-tagged ZFC3H1¹²⁻³³ which revealed a direct interaction between ARS2 and ZFC3H1 mediated by the conserved ZFC3H1¹²⁻³³ motif (Fig. 6e, lanes 4 and 7).”

Discussion (Previously Point 27):

This part now reads as follows (page 14, lines 485-488):

“The Red1 EDGEI motif is conserved in human hZFC3H1¹⁴, a subunit of the PAXT complex. We show that the structure of the C-terminal leg interacting surface is conserved between *S. pombe* and human ARS2. Correspondingly, we also show that hARS2 and hZFC3H1 interact and the E23R mutation in hZFC3H1 (equivalent to E32R in Red1) is sufficient to disrupt the complex.”

Point 2:

Point 6:

Page 8, lines 275 – 281: I suggest to complement these findings with a PISA server analysis of the interface (or parts of it) to maybe obtain more information about physiological relevance.

We now including a sentence with the PISA server analysis (page 8, lines 252-254):
“Analysis of the extensive dimer interface with PDBePISA server³⁶ at the EBI revealed that it buries 4946 Å² on each Ars2 protomer, including 37 hydrogen bond and 18 salt bridge contacts between the two molecules. “

The PISA server also scores the interfaces for their physiological relevance. Maybe this should be included, too. These values given don't tell much, except that the interface is extended.

As suggested, we now also included the Complex Formation Significance Score (CSS) that assesses the relevance of the interface for complex formation (page 7, lines 246-250):

“Analysis of the extensive dimer interface with PDBePISA server³⁶ at the EBI revealed that it buries 4946 Å² on each Ars2 protomer, including 37 hydrogen bond and 18 salt bridge contacts between the two molecules and has a Complex Formation Significance Score (CSS) of 1, indicating that the interface plays an essential role in the dimeric complex formation.”

Point 3:

Point 8:

The abstract is somehow missing a rational why this study was conducted.

The word count of the abstract is limited to 150 words. We think that the rational for conducting this study is explained in the last sentence:

“Our results, combining structures of three Red1 interfaces with *in vivo* studies, provide mechanistic insights into conserved features of MTREC/PAXT architecture.”

What I meant is that there is no scientific question why this study was conducted. E.g. “Inner workings of the PAXT complex still miss sufficient detail.” or alike.

The abstract was now modified to include such a sentence:

“To eliminate specific or aberrant transcripts, Eukaryotes use nuclear RNA-targeting complexes that deliver them to the exosome for degradation. *S. pombe* MTREC complex, and its human counterpart PAXT, are key players in this mechanism but inner workings of these assemblies are not understood in sufficient detail. Here, we present an NMR structure of an MTREC scaffold protein Red1 helix-turn-helix domain bound to the Iss10 N-terminus and show this interaction is required for proper cellular growth and meiotic mRNA degradation. We also report a crystal structure of a Red1-Ars2 complex that explains mutually exclusive interactions of hARS2 with various “ED/EGEI/L” motif-possessing RNA regulators including hZFC3H1 of PAXT, hFLASH or hNCBP3. Finally, we show that both Red1 and hZFC3H1 homo-dimerize via their coiled-coil regions indicating that MTREC/PAXT likely function as dimers. Our results, combining structures of three Red1 interfaces with *in vivo* studies, provide mechanistic insights into conserved features of MTREC/PAXT architecture.”

Point 4:

Point 28:

The discussion may benefit from an overview figure that goes maybe even beyond fig 7f wr to complexity, e.g. extending it towards a hypothetical PAXT assembly.

We now included an extra panel, showing also the hypothetical PAXT complex model (Figure 8b). The *S. pombe* MTREC model was moved to Figure 8a.

Dobrev et al. did not validate the hypothesis, based on conservation of interacting residues, that in human MTR4 interact with ZFC3H1 with equivalent regions as in *C. thermophilum*. We now used AlphaFold2 to predict the human complex structure revealing the interaction of the expected the MTR4-binding region of ZFC3H1 which in addition also packs against a small downstream helical domain. Since, it is likely that the complete MTR4-binding domain spans both regions, we included these models in a new Supplementary Figure 15. An equivalent domain was predicted also for *S. pombe* Red1 (AF-Q9UTR8, residues 575-703).

Supplementary figure 15 is not referenced in the main text. And I would not use the term ‘revealing’ for an AlphaFold2 model, in particular, when it comes to side chain interactions.. It is merely an indication

We thank the Reviewer for pointing this out.

In fact, the Figure 8b legend did not contain the word “revealing”, but we further modified it as follows (page 28, lines 957-961):

“AlphaFold2³⁵ structure prediction of the human complex structure (Supplementary Fig. 15) indicates that that MTR4-binding region of ZFC3H1 might pack against a small helical domain. It is thus likely, that the MTR4-binding domain of ZFC3H1 covers residues 1121-1292. Similarly, in the AlphaFold2 prediction of Red1 (AF-Q9UTR8), this domain spans residues 575-703.”

The reference to Supplementary Figure 15 is in a legend of Figure 8. This supplementary figure is only supporting the Figure 8b but is not discussed in the main text. We now also added the reference to Sup Fig. 15 to

the last sentence of the Results section, as follows (page 12, lines 420-421):

“The dimeric architecture might thus be a conserved feature among all these RNA targeting complexes (Fig. 8b, Supplementary Fig. 15).”

Reviewer #3

The revised version of the manuscript did not change initial concerns that were raised by me and one of the other reviewers: previously published data has not been acknowledged appropriately. In addition, I feel that the paper is in part not focussed enough. Three different topics or interactions (Red1-Iss10, Red1-Ars2 and Red1-Red1) that are presented are clearly related, but there are no real connections between the three parts. As a result, the paper fails a clear message. Finally, the solved structures are very standard and do not provide exciting or state-of-the-art insights. I suggest publishing the work in a more focused journal, e.g. NAR or RNA. The impact of the manuscript appears too limited to warrant publication in Nat. Com.

We are very sorry that the previously published data of Dobrev *et al.*, some of which we had also obtained, is still not adequately acknowledged. We sincerely apologize for this and have now made a further effort to completely rectify this failing, based on the reviewer's comments. We trust that the referee will now consider it done a satisfactory way.

We agree with the reviewer that the structures we report are not necessarily as exciting and revealing as some of the large cryo-EM structures now frequently determined. Nevertheless, an important aspect to consider is the maturity and complexity of respective research fields. For example, the core of the related NEXT complex consists of three proteins only. The initial biochemical and structural analyses focused on sub-complexes (Falk *et al.*, Nat. Comm., 2016, Puno *et al.*, PNAS 2018) but almost certainly were essential groundwork that paved the way towards the cryo-EM structure of the core complex published last week (Puno and Lima, Cell 2022, Gerlach *et al.*, Moll Cell 2022). The MTREC and PAXT are larger complexes containing more subunits and their architecture is currently poorly understood. At this stage, the available data on MTREC and PAXT are too limited to target the full complex at the structural level, and the structures of Dobrev *et al.* and ours, while describing only sub-complexes, have the potential to contribute towards the eventual structural characterization of the whole assembly, as initial biochemical and structural analyses recently did for the NEXT complex.

In our view, the three aspects of Red1 that we address are clearly connected and provide new insights into how Red1/ZFC3H1 orchestrates the overall MTREC/PAXT architecture. In fact, the multiple features of Red1/ZFC3H1 interactions characterized in the study broaden the impact of the work. In addition to researchers working on RNA targeting complexes, our study is also informative for the meiotic RNA degradation field studying Mmi1/Iss10 and the regulatory function of the cap-binding complex. Indeed, CBC-ARS2 is likely involved in several layers of mutually exclusive interactions with other RNA regulators to determine the RNAs fate. One such interface has been characterized on CBC (Schulze and Cusack, Nat. Comm. 2017). Our work now provides structural insights into another one on ARS2, complementing previous biochemical studies (Schulze *et al.*, Nat. Comm. 2018, Dobrev *et al.*, Nat. Comm. 2021). In summary, we believe that our work is complementary to the studies of Dobrev *et al.*, 2021 and Schulze *et al.* 2018, and together these constitute an important step towards better understanding the MTREC/PAXT structure and function and the role of CBC-ARS2 in RNA maturation.

My conclusion is based on the following remarks:

Point 1:

The NMR structure is new. Considering the very small size of this complex it can not be considered technically challenging though. I wonder what an alpha-fold prediction would give. The authors did quite some alpha-fold and are thus able to address this.

With all due respect, the technical challenge of obtaining a structure is not necessarily related to, or even relevant to, its significance. Despite its modest size the Red1-Iss10 complex resisted crystallization but could be solved by NMR. It was not trivial to obtain this structure. The structure shows with high detail and confidence the interaction interface. We now understand atomic details of the interaction and could use a structure-based mutant in functional experiments to better understand the role of this interaction *in vivo*.

The question about AlphaFold2 (AF) predictions is an interesting and important one. Indeed, the AF ability to

predict protein structures, often with high accuracy, is a game changer in the field of structural biology. However, we share the current consensus that whereas AF modeling can be of great help for hypothesis generation, construct design and crystal structure phasing, the structure predictions themselves are just predictions that are certainly not perfect (for many possible reasons) and therefore need independent validation. Experimental structure determination is an obvious way to do this. Actually, the Red1-Iss10 complex structure was determined before AF became available, so the question in this case is moot. However, we can state that the AF model seems overall quite correct, but locally differences occur.

Point 2:

Line 143-144: ‘Noticeably, similar domain boundaries of the two proteins required for the interaction were recently defined by Y2H assays’. Why is this noticeable? It is just a repeat of previous experiments. In case the authors learned anything new they should highlight that. In case nothing new is learned they should just clearly state that they repeated the experiments that have been published before. (See top green arrow Fig 2C of ref 14).

We agree with the reviewer’s comment that the previously published work was not acknowledged properly. However, we would like to clarify that we did not repeat previously published results. We obtained all the results independently, but as we performed also additional structural and functional analyses, this study took longer to complete.

This section was now modified as follows (page 4, lines 135-140):

“A Y2H and GST pull-down analysis recently revealed that Red1¹⁸⁷⁻²³⁶ and Iss10¹⁻⁵¹ represent the interacting regions of the complex Red1 forms with Iss10¹⁴. Our characterization of the complex using Y2H, Strep-tag pull-down and limited proteolysis experiments identified equivalent fragments Red1¹⁹²⁻²³⁶ and Iss10¹⁻⁴⁵ (Supplementary Fig. 1c-e). Multi angle laser light scattering (MALLS) showed that these constructs form a homogenous complex sample with a 1:1 stoichiometry (Supplementary Fig. 1f).”

Point 3:

Line 157-158: “While Red1 DNA binding has not been reported and no unspecific DNA binding was observed during Red1 purification, DNA interaction cannot be excluded.”. It is up to the authors to exclude (or at least test) this, e.g. using NMR titration experiments.

This sentence was missing the word “specific”. It is now corrected (page 5, lines 151-152):

“While Red1 DNA binding has not been reported and no unspecific DNA binding was observed during Red1 purification, specific DNA interaction cannot be excluded.”

As we wrote in the results section, DNA binding of Red1 has never been reported and our structure suggests it is unlikely. We also do not see DNA co-purification during Red1 purification that would be an indication of unspecific DNA binding. Following the direction of possible interaction between Red1 and DNA may or may not yield interesting results, but as discussed above, our study already covers three different aspects of Red1. We think that identifying possible specific target DNA sequence or confirming and characterizing or disproving non-specific binding with Red1 is out of the scope and interest of this study.

Point 4:

Line 174-175: “identifying Red1 mutations that could disrupt its interaction with Iss10 without affecting the Red1 HTH fold.”. The authors should show that their mutations (L205R and F215R) indeed to not destabilize the HTH fold. They should e.g. provide NMR spectra of the mutant proteins. In case the HTH fold is destabilized by the mutations, all in vivo effects could be due to an unfolded HTH domain and not be due to the disrupted interaction between the 2 proteins.

We agree with the Reviewer that it is important to validate that structure-based mutations do not destabilize the mutated protein. In the first round of revision, Reviewer 2 requested showing gel-filtration profile of the Red1 L205R mutant compared to the WT, and we provided those results. However, it is true that the Red1 fragment in question was fused to the MBP tag, which could mask the possible impact of the mutation on the domain. We now purified Red1(192-236) and the two mutants without the MBP tag. 1D NMR analysis shows that Red1(192-236) in isolation (without Iss10) is unfolded. This is common for the interactions of two peptides in small domains and one of many examples can be seen in Figure 1 in Moreno-Morcillo et al (2011) (PMID: 21481776). Importantly, the 1D NMR spectra together with gel filtrations of WT and mutated Red1(192-236) shows that L205R and F215R mutants are similar in behaviour to WT Red1¹⁹²⁻²³⁶ being soluble with no sign of aggregation or higher order oligomerization (as evident by the narrow NMR peak linewidths). The additional experiments were added as Supplementary Figure 2g and h, replacing the previous gel filtrations of MBP fused proteins.

In addition, we would like mention that our *in vivo* results show that the Red1 L205R has a RNA degradation phenotype similar to the cells lacking Iss10, and which is weaker compared to the cells lacking Red1, which provides another indication that other Red1 functions were not compromised by the L205R mutation.

This section now reads as follows (page 6, lines 175-183):

“To assess possible impact of the L205R and F215R mutations on the Red1 HTH domain, we produced Red1¹⁹²⁻²³⁶, but 1D NMR analysis revealed that in absence of Iss10, the WT and mutated variants are unfolded (Supplementary Fig. 2g). Together with size exclusion chromatography, the 1D NMR spectra confirmed that both L205R and F215R mutants are similar in behaviour to WT Red1¹⁹²⁻²³⁶ being soluble with no sign of aggregation or higher order oligomerization (Supplementary Fig. 2g,h). We then confirmed the essential role of L205 and F215 for the interaction with Iss10 in the context of full-length proteins using an Y2H assay (Fig. 2j). Ultimately, the L205R mutation was selected for *in vivo* studies (see below).”

Point 5:

Line 223-224: “To further characterize the interaction between Ars2 and this Red1 motif, which has also recently been shown to bind Ars2 by Dobrev et al.14, “. The authors still fail to acknowledge previous data appropriately. In ref 14 the interaction between Ars2 and Red1 has been studied in quite some detail. The findings presented here just recapitulate published findings. The ignorance of the authors towards published data from others is further obvious from the statement “further supporting the hypothesis that Red1 region spanning residues 20-40 is a major determinant of the Ars2 binding.”. Ref 14 clearly identified this region and even presents point mutations that disrupt the interaction Red1-Ars2 interaction.

We agree with the Reviewer and we modified this section accordingly (page 7, lines 212-219):

“The interaction between Ars2 and Red1 has recently been shown, by Y2H and pull-down assays, to be mediated by a short conserved “EDGEI” motif of Red1¹⁴. Using Y2H and gel filtration chromatography we confirmed the implication of the EDGEI motif in the interaction with full-length Ars2, identifying a slightly longer construct Red1²⁰⁻⁴⁰ (Fig. 4a, Supplementary Fig. 4). To further characterize the interaction between Ars2 and this Red1 motif, we used isothermal titration calorimetry (ITC). Both Red1¹⁻⁵⁴ and Red1²⁰⁻⁴⁰ (with and without MBP tag) bound Ars2 with an equivalent dissociation constant (K_d) of about 5 μ M (Fig. 4b, Supplementary Fig. 5a,b), confirming that additional sequences surrounding EDGEI motif do not enhance the binding affinity for Ars2.”

Point 6:

Line 244-246: “ The structure was determined by molecular replacement using the AlphaFold2 Ars2 model (AF-O94326) and refined to an Rfree of 30.2% and an Rworkof 24.5% (Supplementary Table 2).”. The fact that the alpha-fold model can be used for molecular replacement raises the question how different the alpha-fold model is from the determined structure. The authors should report an overlay of the model and the structure. In case both are highly similar then I would say that it is fair to ask what the added value of the structure is. Clearly, the structure is a complex, but maybe alpha-fold is also able to predict the structure of the complex well.

As discussed above, the AF models are still associated with uncertainty of not being confident about which model or which part of the model is correct and which is not. We agree with the reviewer that the AF model and the final experimental structure will likely be similar since the predicted model was successfully used for molecular replacement. Nevertheless, even if they were identical, the experimental model would still have an essential value, since without it one could not be fully confident that the prediction is correct. This situation might change in the future, if AF further evolves and the structural biology community agrees on some quality measures that would eliminate the need for experimental structures. In this case, at least at the time of determining the structure of the complex, AF was able to correctly place only the Glu35 of Red1, the other charged residues of the motif were predicted wrongly. Given that we have an experimentally determined structure, which was deposited to the PDB satisfying their strict quality control measures and indicators (currently not existing for submitting and sharing AF models of complexes), we do not think that adding an AF prediction model of this complex would add extra value to the manuscript.

Point 7:

The “Dimerization of Red1 C-terminus” does not provide any (functional) insights. A mutant that prevents the dimerization is identified, but the role of dimerization (if any exists) remains unclear.

We were also a little disappointed that the mutation identified to disrupt the Red1 dimer *in vitro* had no

detectable impact *in vivo*. However, as discussed in Discussion, the dimerization partially persists even in the mutated cells, which might be responsible for the lack of apparent phenotype. Whether Red1 remains dimeric via other subunit or RNA remains to be determined.

Despite this, this part of the study still reports several novel and interesting observations. It is the first report of the MTREC dimerization. We identified a determinant of the dimerization (coiled-coil), overall validated the model predicted by AF, showed that structural model-based mutations abolished the binding *in vitro*, as well as we demonstrated Red1 dimerization *in vivo*. We also predicted that the PAXT and NEXT dimerize as well. Interestingly, the very recent studies from C. Lima and E. Conti labs, published last week in Cell (PMID: 35688134) and Molecular Cell (PMID: 35688157), respectively, report a cryo-EM structure of the dimeric core of the NEXT complex, showing that NEXT dimerization enhances the helicase activity of MTR4 *in vitro*, while the observed effect of RNA targets *in vivo* is rather modest. Our results thus seem timely and, importantly, show that dimerization is a common feature between these RNA targeting complexes.